# A circuit from the ventral subiculum to anterior hypothalamic nucleus GABAergic neurons essential for anxiety-like behavioral avoidance

Jing-Jing Yan[1], Xiao-Jing Ding[1,10], Ting He[2,3,10], Ai-Xiao Chen[1], Wen Zhang[1], Zi-Xian Yu[1,4], Xin-Yu Cheng[5,6], Chuan-Yao Wei[1], Qiao-Dan Hu[1], Xiao-Yao Liu[1], Yan-Li Zhang[1], Mengge He[2,3], Zhi-Yong Xie[5], Xi Zha[1], Chun Xu [1,7], Peng Cao [5,8], Haohong Li[9] & Xiao-Hong Xu [1,7] ✉

Behavioral observations suggest a connection between anxiety and predator defense, but the underlying neural mechanisms remain unclear. Here we examine the role of the anterior hypothalamic nucleus (AHN), a node in the predator defense network, in anxiety-like behaviors. By in vivo recordings in male mice, we find that activity of AHN GABAergic (AHN$^{Vgat+}$) neurons shows individually stable increases when animals approach unfamiliar objects in an open field (OF) or when they explore the open-arm of an elevated plus-maze (EPM). Moreover, object-evoked AHN activity overlap with predator cue responses and correlate with the object and open-arm avoidance. Crucially, exploration-triggered optogenetic inhibition of AHN$^{Vgat+}$ neurons reduces object and open-arm avoidance. Furthermore, retrograde viral tracing identifies the ventral subiculum (vSub) of the hippocampal formation as a significant input to AHN$^{Vgat+}$ neurons in driving avoidance behaviors in anxiogenic situations. Thus, convergent activation of AHN$^{Vgat+}$ neurons serves as a shared mechanism between anxiety and predator defense to promote behavioral avoidance.

Anxiety represents an emotional state of apprehension about remote, potential, unpredictable, or ill-defined threats[1–4]. It keeps individuals vigilant about potential harms and prepared for safety measures[4,5]. Using behavioral tests that exploit the "approach-avoidance" conflict[6],

such as the open field test and the elevated plus-maze (EPM), previous studies have identified brain areas that work in concert to regulate approach-avoidance behaviors[7–10]. Some of these brain regions, such as ventral CA1 (vCA1) of the hippocampus, the lateral septum nuclei

[1]Institute of Neuroscience, State Key Laboratory of Neuroscience, CAS Center for Excellence in Brain Science and Intelligence Technology, Chinese Academy of Sciences, 200031 Shanghai, China. [2]Britton Chance Center for Biomedical Photonics, Wuhan National Laboratory for Optoelectronics, Huazhong University of Science and Technology, 430074 Wuhan, Hubei, China. [3]MoE Key Laboratory for Biomedical Photonics, Collaborative Innovation Center for Biomedical Engineering, School of Engineering Sciences, Huazhong University of Science and Technology, 430074 Wuhan, Hubei, China. [4]University of Chinese Academy of Sciences, 100049 Beijing, China. [5]National Institute of Biological Sciences, 102206 Beijing, China. [6]Graduate School of Peking Union Medical College, Chinese Academy of Medical Sciences, Beijing, China. [7]Shanghai Center for Brain Science and Brain-Inspired Intelligence Technology, 200031 Shanghai, China. [8]Tsinghua Institute of Multidisciplinary Biomedical Research, Tsinghua University, Beijing 100084, China. [9]The MOE Frontier Research Center of Brain & Brain machine Integration, Zhejiang University School of Brain Science and Brain Medicine, 310058 Hangzhou Zhejiangode, China. [10]These authors contributed equally: Xiao-Jing Ding, Ting He. ✉e-mail: xiaohong.xu@ion.ac.cn

(LS), and the bed nucleus of the stria terminalis (BNST), modulate approach-avoidance behaviors through projections to specific hypothalamic nuclei[11–13], particularly the lateral hypothalamus (LH).

Intriguingly, brief predator encounters elevate anxiety levels in species ranging from flatworms to fish, rodents, and primates[14–17], pointing to an evolutionarily conserved mechanism connecting predator defense to anxiety[18–21]. Indeed, the Predator Imminence Continuum theory proposes three modes of defensive behaviors organized along a predator's perceived imminence continuum (spatial/temporal distance or probability of contact)[10,22], of which anxiety corresponds to the "pre-encounter" mode. Likewise, fear and panic map to the "post-encounter" and "circa-strike" modes[10,22]. Supporting this view, animals selectively bred for high anxiety traits show increased avoidance of predator cues[23]. Additionally, anti-anxiety drug treatments diminish predator cue avoidance in normal animals[24,25]. Furthermore, human subjects having high trait anxiety were more inclined to make early escapes in response to slow but not fast-approaching predator figures in a computer game[19]. Despite these behavioral and theoretical works, the neural circuit mechanism accounting for the convergence between anxiety and predator defense remains poorly understood.

The present study focused on the anterior hypothalamic nucleus (AHN), which reciprocally connects with the ventromedial hypothalamus (VMH) and the dorsal premammillary nucleus of the hypothalamus (PMd) to form the hypothalamus predator defense network in rodents[26,27]. Predator cues activate VMH and PMd[27–32]. Optogenetic activation of VMH neurons or their projections to AHN is sufficient to drive avoidance in mice[33]; however, loss-of-function experiments have not demonstrated a clear role for AHN in predator defense. By contrast, both gain- and loss-of-function experiments and studies of genetically defined populations have shown that VMH and PMd regulate essential post-encounter defense behaviors such as freezing and flight[34–38]. Interestingly, anti-anxiety drug treatment reduces predator cue-induced c-Fos signals in AHN but not VMH[25]. In addition, LS neurons that express type 2 corticotropin-releasing factor receptor (Crfr2) enhance stress-induced anxiety behaviors and cortisol release through projections to AHN[12]. Based on these results, we focused on AHN neurons as a potential convergence site for neural circuits linking predator defense and anxiety.

We first found that the activity of AHN GABAergic neurons (AHN^Vgat+) strongly correlated with the mouse avoidance behaviors in two anxiety tests, with each mouse exhibiting consistent and individual-specific changes in AHN^Vgat+ activity. Moreover, anxiety-related AHN neuronal ensembles overlapped with those evoked by a predator cue. Furthermore, optogenetic inhibition of AHN^Vgat+ neurons during exploration reduced subsequent avoidance behaviors. Using pseudorabies virus retrograde tracing, we further identified the ventral subiculum (vSub) of the hippocampal formation as a significant input to AHN^Vgat+ neurons in driving avoidance behaviors. Together, these results point to AHN^Vgat+ neurons as a point of convergence between anxiety and predator defense. Thus, AHN^Vgat+ neurons promote characteristic anxiety-like avoidance behavior by incorporating inputs from the hippocampal formation.

## Results
### Strong temporal correlation of AHN^Vgat+ neuron activity with object-evoked avoidance in a modified open field paradigm
Center avoidance and peripheral preference in an open field test are behavioral parameters that indicate rodent anxiety levels[6]. As previously reported, we found that wildtype male mice spent more time around the wall in an open field test (Fig. 1a). However, by introducing an unfamiliar object (a battery) to the center of the open field -10 min after a mouse freely explored the arena, we found that this procedure led to more substantial center avoidance and peripheral preference

(Fig. 1a), indicating that an unfamiliar object elevates the anxiety level of the animal. Such behavioral changes were not observed in control animals that were allowed to explore the open field continuously for 20 min, with the experimenter's hand interruption briefly without placing the object (Fig. 1b). Thus, object-evoked behavioral changes were unlikely caused by fatigue, habituation, or human interference.

Although a still object differed greatly from a real predator, both produced a thigmotaxis phenotype in the open field test[34]. We, therefore, inquired whether the hypothalamus predator defense circuit, particularly AHN, was engaged during object-induced center avoidance (Fig. 1c). First, our in situ mapping of mRNAs of vesicular transporters for GABA and glutamate (Vgat and Vglut2) showed that, among all Vgat+ and Vglut2+ neurons in AHN, the vast majority (83.0 ± 5.7%, $n = 3$ mice) expressed Vgat (Fig. 1d). We thus used the Vgat-IRES-Cre line to target AHN Vgat+ (AHN^Vgat+) neurons. We independently validated the fidelity of this mouse line by injecting adeno-associated virus (AAVs) encoding Cre-inducible EYFP into AHN, in which we found 98.9 ± 0.7% of GFP+ neurons expressed Vgat (Supplementary Fig. 1a, $n = 3$ mice).

To monitor the activity of AHN^Vgat+ neurons, we injected AAVs encoding Cre-inducible GCaMP6s, or EYFP as the control, into AHN of Vgat-IRES-Cre mice and implanted an optic fiber above the injection site (Fig. 1e). These procedures did not result in apparent changes in object-evoked avoidance behavior in the open field (before vs. after object introduction, peripheral zone time, 385.8 ± 10.8 s vs. 500.7 ± 16.3 s, Wilcoxon matched-pairs signed rank test, $p < 1 \times 10^{-4}$, $n = 22$ mice). Before object introduction, GCaMP6s signals of AHN^Vgat+ neurons were not significantly modulated by the location of the animal in the open field (Fig. 1f). Remarkably, after object introduction, GCaMP6s signals of AHN^Vgat+ neurons elevated considerably, with the most dramatic increase observed when the mouse arrived at the open field center zone (Fig. 1f). Notably, such fluorescent signal changes were not observed in control animals that expressed EYFP in AHN^Vgat+ neurons (Supplementary Fig. 1b–d), indicating that changes in GCaMP6s signals were unlikely caused by motion artifacts.

Moreover, we found that AHN^Vgat+ GCaMP6s signals tracked with the animal's distance relative to the object with little adaptation as the trial went on (Fig. 1g). Specifically, AHN^Vgat+ GCaMP6s signals ramped up as the animal approached the object and down as it retreated to the peripheral zone (Fig. 1h, i). For individual trials, the overall temporal dynamics of AHN^Vgat+ fluorescence signals strongly correlated with "approach-retreat" bouts in GCaMP6s mice with an average correlation coefficient ($r$) of 0.28 ± 0.05 ($n = 14$ mice), significantly higher than that of EYFP control mice ($r = -0.03 ± 0.01$, $n = 8$ mice; Unpaired $t$-test, $p < 1 \times 10^{-4}$). Furthermore, the peak value of AHN^Vgat+ GCaMP6s signals at the end of a center approach, a turning point before the retreat, positively correlated with the latency to initiate the next approach (Fig. 1j, $r = 0.28$, $p < 1 \times 10^{-4}$), suggesting that close encounter with the object elevated the anxiety. Along the same line, the average "turning point" AHN^Vgat+ GCaMP6s signals in a trial significantly correlated with the total duration the animal spent in the peripheral zone away from the object (Fig. 1k, $r^2 = 0.28$, $p = 0.0055$).

As a comparison study, we placed an object in the mouse's home cage for three days for familiarization and then performed the open field test with the familiarized object in the center (Supplementary Fig. 1e). Interestingly, despite intense object investigation, AHN^Vgat+ GCaMP6s signals did not change during approach or retreat (Supplementary Fig. 1f–h). In addition, no changes in AHN^Vgat+ GCaMP6s signals were observed when the mouse investigated, sniffed, or mounted a novel female mouse introduced to its homecage (Supplementary Fig. 1i–l), demonstrating that AHN^Vgat+ neuron activity does not simply reflect stimulus novelty, exploratory actions or social activity. Together, these results show a robust temporal correlation between AHN^Vgat+ neuron activity and anxiety-like avoidance behavior.

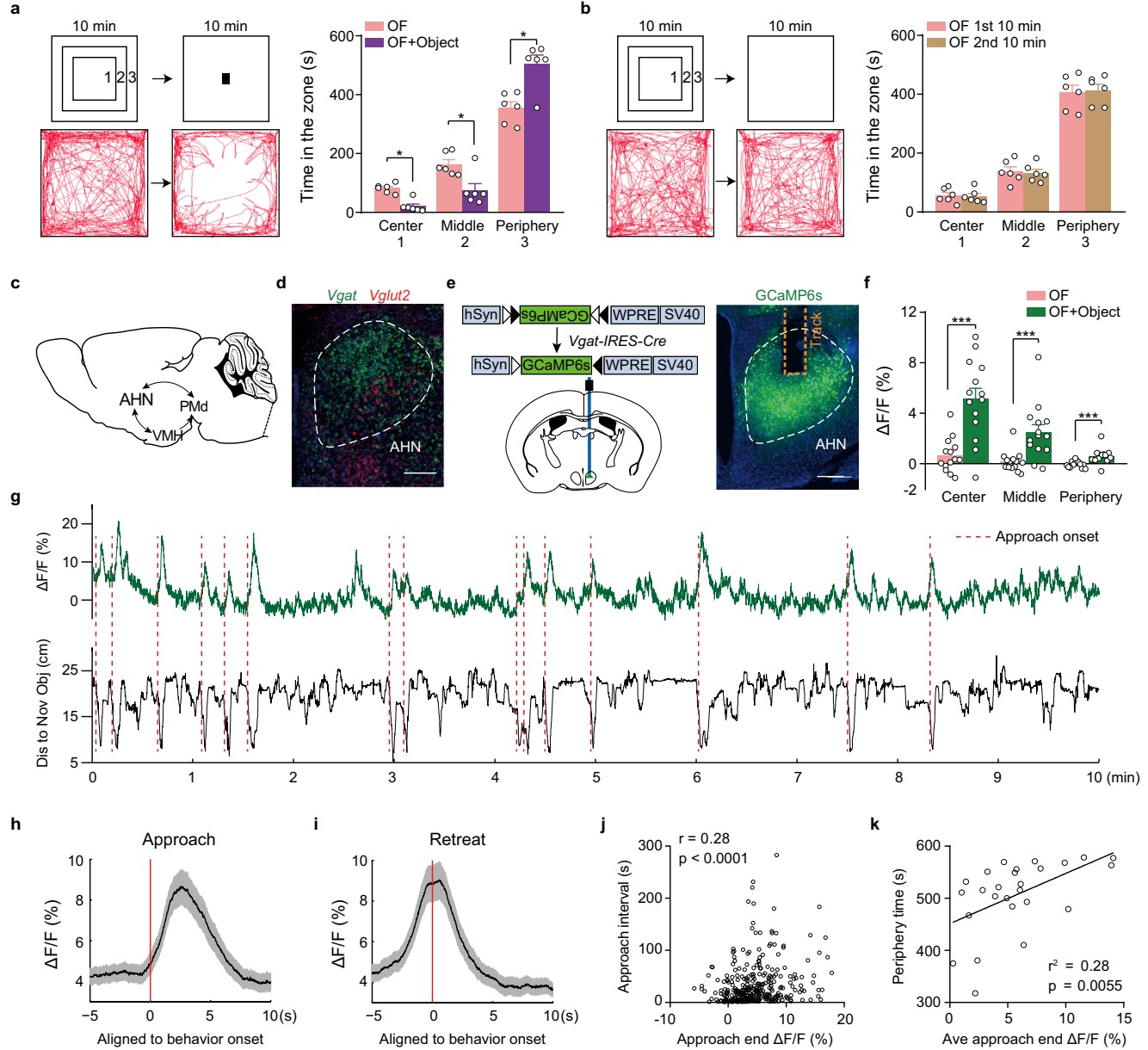

Fig. 1 | Strong temporal correlation of AHN^Vgat+ neuron activity with object-evoked avoidance behavior in a modified open field paradigm. a We modified the open field test with the introduction of an unfamiliar object 10 min after the initial exploration. "1", "2", and "3" denote the "center", "middle", and "peripheral" zone of the open field. The example trajectory (bottom) and the quantification (right) show that animals spent more time in the peripheral zone away from the center after object introduction. $n = 6$ mice. Wilcoxon matched-pairs signed rank test, center, $p = 0.03$, middle, $p = 0.03$, peripheral, $p = 0.03$. b In control assays, animals were allowed to explore the open field for 20 min. The example trajectory and the quantification (right) show similar time spent in all three zones in the first and second 10 min of the open field test. $n = 6$ mice. Two-tailed paired t test. c Schematic illustration of the "hypothalamus predator defense circuit". d A representative image showing the fluorescent in situ signals of *Vgat* and *Vglut2* mRNA in AHN. Scale bar, 200 μm. e Left, the strategy to monitor GCaMP6s signals in AHN^Vgat+ neurons. Right, a representative image showing restricted GCaMP6s

expression in AHN. Scale bar, 200 μm. f Average $\Delta F/F$ values detected in the "center", "middle" and "periphery" zone before and after object introduction in GCaMP6s animals. $n = 14$ mice. Center, two-tailed paired t test, center, $p = 0.0003$; middle and peripheral, Wilcoxon matched-pairs signed test, $p = 0.0004$ and $p = 0.0009$, respectively. g A representative trace of $\Delta F/F$ signals (green, top) aligned to the relative distance (black, bottom) between a GCaMP6s animal and the object. Red dashed lines denote the onset of approach bouts. h, i Average $\Delta F/F$ values of GCaMP6s signal aligned to approach (h) or retreat onset (i) at the time "0". Shades indicate the SEM. j Correlation between the GCaMP6s $\Delta F/F$ value at the end of an approach and the latency to initiate the following approach. $n = 351$ bouts from 14 mice. Spearman correlation. k Correlation between average approach-end GCaMP6s $\Delta F/F$ value and the time animals spent in the periphery zone. $n = 26$ trials from 14 mice. Pearson's correlation. *$p < 0.05$; ***$p < 0.001$. Data are presented as mean values ±SEM.

## Object-evoked AHN^Vgat+ activity shows individual specificity

To investigate whether any specific features of the object used (a battery) is responsible for evoking AHN^Vgat+ activity, we performed a new set of experiments using three other alternative items, an acrylic cuboid cube, a toy airplane, and a metal paper clip, in addition to the battery, as the unfamiliar object (Supplementary Fig. 2a). We

individually presented these four objects on separate testing days in a pseudo-randomized order (Supplementary Fig. 2b). We found that all objects drove the tested animals to spend more time in the peripheral zone after being introduced to the open field (Supplementary Fig. 2c). Furthermore, we found a similar temporal correlation of ramping AHN^Vgat+ GCaMP6s signals with approach-retreat bout and with the

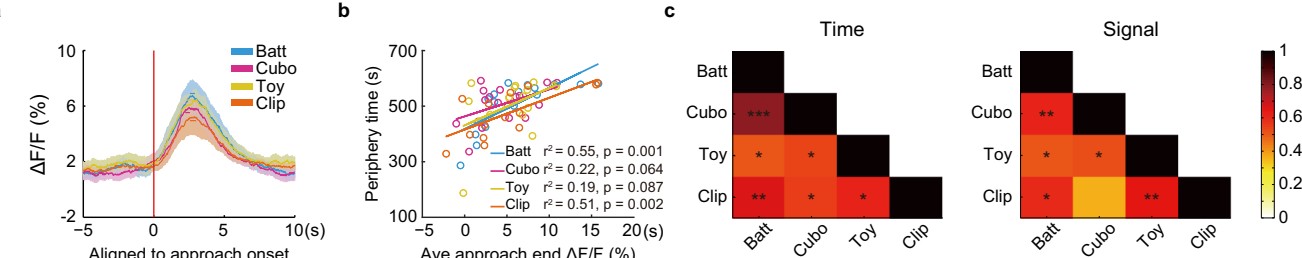

**Fig. 2 | Object-evoked AHN$^{Vgat+}$ activity shows individual specificity. a** Average $\Delta F/F$ signals ± SEM (shades) aligned to approach onset toward different unfamiliar objects: battery (Batt), cuboid cube (Cubo), toy airplane (Toy), and paper clip (Clip). Horizontal lines around the peak values show the average retreat onset ±SEM. **b** Correlations between average approach-end $\Delta F/F$ value and the time spent in the open field periphery zone after the introduction of different objects. $n = 16$ mice. Pearson correlation. **c** Pair-wise correlations between the time spent in the periphery zone (left) and the average approach-end $\Delta F/F$ value (right) across the four object conditions. The heat map (scale on the right) represents the correlation co-efficiency (r) value with the $p$ values, indicated by stars, depicted in each cell for each pair. *$p < 0.05$; **$p < 0.01$; ***$p < 0.001$.

time spent in the peripheral zone for all four objects (Fig. 2a, b). Notably, the AHN$^{Vgat+}$ GCaMP6s signals and avoidance behavior evoked by the unfamiliar object were variable among different mice, yet the same mouse showed highly consistent responses towards different objects. Further pair-wise analysis showed a strong correlation of data between trials of two different objects, for the average turning point GCaMP6s signals and the time spent in the peripheral zone (Fig. 2c). This individual specificity further supports the notion that elevated AHN$^{Vgat+}$ neuron activity underlies object-evoked avoidance behavior.

## Inhibiting object-evoked AHN$^{Vgat+}$ activity reduces avoidance

We next examined whether inhibiting AHN$^{Vgat+}$ neuron activity during the object approach could abolish object-induced increases in anxiety and avoidance behavior. To this end, we bilaterally injected AAVs encoding Cre-inducible GtACR1, or EYFP as the control, into AHN of *Vgat-IRES-Cre* male mice (Fig. 3a) and implanted an optic fiber 300–500 μm above each injection site (Fig. 3b). We used ex vivo patch-clamp recordings to confirm that pulses of blue light (473 nm, 20 ms, 20 Hz) effectively and reversibly silenced GtACR1-expressing AHN$^{Vgat+}$ neurons (Fig. 3c, d). By analyzing fiber-photometry recorded animals (Fig. 1), we found that the starting point for approach bouts toward the object was mostly located within the peripheral zone (Fig. 3e). Therefore, we delivered light pulses whenever the mouse left the peripheral zone after the object introduction to inhibit AHN$^{Vgat+}$ neuron activity during the object approach (Fig. 3e). These light pulses had no effect in control EYFP mice but completely abolished the object avoidance in GtACR1 mice (Fig. 3f–g). Furthermore, freezing behaviors were also diminished in GtACR1 mice during light stimulation (Supplementary Fig. 3a). By comparison, light delivery had no effects on jumping, stretch-attend postures (SAP), or locomotion (Supplementary Fig. 3b–d).

Notably, such behavioral effects were specific to inhibiting AHN$^{Vgat+}$ neurons and were not found when we targeted AHN *Vglut2* + (AHN$^{Vglut2+}$) neurons using the *Vglut2-IRES-Cre* line[39]. Even though AHN$^{Vglut2+}$ neurons also showed object-evoked activity increases in the open field center zone (Supplementary Fig. 4a, b), GtACR1-mediated inhibition of AHN$^{Vglut2+}$ neurons increased, rather than decreased, the time that animals spent in the peripheral zone (Supplementary Fig. 4c, d). These results showed that inhibiting AHN$^{Vglut2+}$ neurons promoted, rather than reduced, object-evoked anxiety. Thus, AHN$^{Vgat+}$ but not AHN$^{Vglut2+}$ neuron activity promoted anxiety-like behavioral avoidance of an object in the open field.

Because the activity of AHN$^{Vgat+}$ neurons climaxed before the retreat, we further inhibited these neurons with more precise temporal control by applying the light pulses when mice arrived at the center zone where the unfamiliar object was placed, and most retreat bouts were initiated (Fig. 3h). For this set of experiments, we recorded baseline behavior for 10 min before and after the introduction of an object (a battery or a cuboid) to the center (Fig. 3i), and the light was then delivered whenever the mouse arrived at the center zone during the next 10 min (Fig. 3i). Remarkably, optogenetic inhibition of AHN$^{Vgat+}$ neurons in the center zone drastically reduced the object avoidance, as shown by more time spent in the center zone and less time in the peripheral zone during the inhibition phase as compared to that during the baseline period (Fig. 3i, j and Supplementary Fig. 3e, f). In addition, light mildly reduced freezing behaviors without significantly affecting jump, stretch-attend postures (SAP), or locomotion in GtACR1 males compared to the controls (Supplementary Fig. 3g–j).

Strikingly, GtACR1 animals exhibited elevated center zone exploration even after the cessation of light, indicating a persistent effect of AHN$^{Vgat+}$ inhibition (Fig. 3i, j and Supplementary Fig. 3e, f). In these experiments, the extended duration the mice spent in the presence of the object (30 min) did not reduce anxiety since EYFP control mice showed no reduction of object avoidance before and after light stimulation (Fig. 3j and Supplementary Fig. 3e, f). Instead, we found that GtACR1 animals spent more time close up to the unfamiliar object during light stimulation (Supplementary Fig. 3k). Some GtACR1 animals even climbed onto the object (Supplementary Fig. 3l), a behavior rarely observed in control males or before light stimulation in GtACR1 males. We suspect these close contacts fastened object familiarization, leading to reduced anxiety/avoidance after light termination.

Crucially, the behavioral effects of optogenetic inhibition in GtACR1 animals could NOT be attributed to light-conditioned place preference (CPP). In a CPP apparatus consisting of two chambers (Supplementary Fig. 5a), EYFP or GtACR1 animals showed no consistent avoidance of either chamber (Supplementary Fig. 5b), indicating that neither is particularly anxiogenic. When we randomly paired light delivery to one of the two chambers (Supplementary Fig. 5c), light did not lead to a preference for the paired chamber in EYFP or GtACR1 animals (Supplementary Fig. 5d), demonstrating that inhibiting AHN$^{Vgat+}$ neurons did not produce a CPP effect. Together, these experiments support that elevated AHN$^{Vgat+}$ neuron activity is necessary for object-induced anxiety and avoidance behavior.

## Overlapping AHN neural ensembles respond to an object and a predator cue

To examine whether AHN$^{Vgat+}$ neurons activated by an object overlaps with those responding to a predator cue (fox urine), we performed the compartmental analysis of temporal activity by fluorescent in situ hybridization (catFISH). This method allowed us to track neural ensembles activated by two stimuli, presented ~30 min apart, via the expression of the cytoplasmic and nuclear *c-Fos* transcript[40]. We sequentially exposed animals to an unfamiliar object in an open field or fox urine in a new cage, or two different objects (Fig. 4a). We then stained AHN sections for *Vgat*, *c-Fos* exon (cytoplasmic), and *c-Fos* intron (nuclear) expression (Fig. 4b). In all groups, the first stimuli

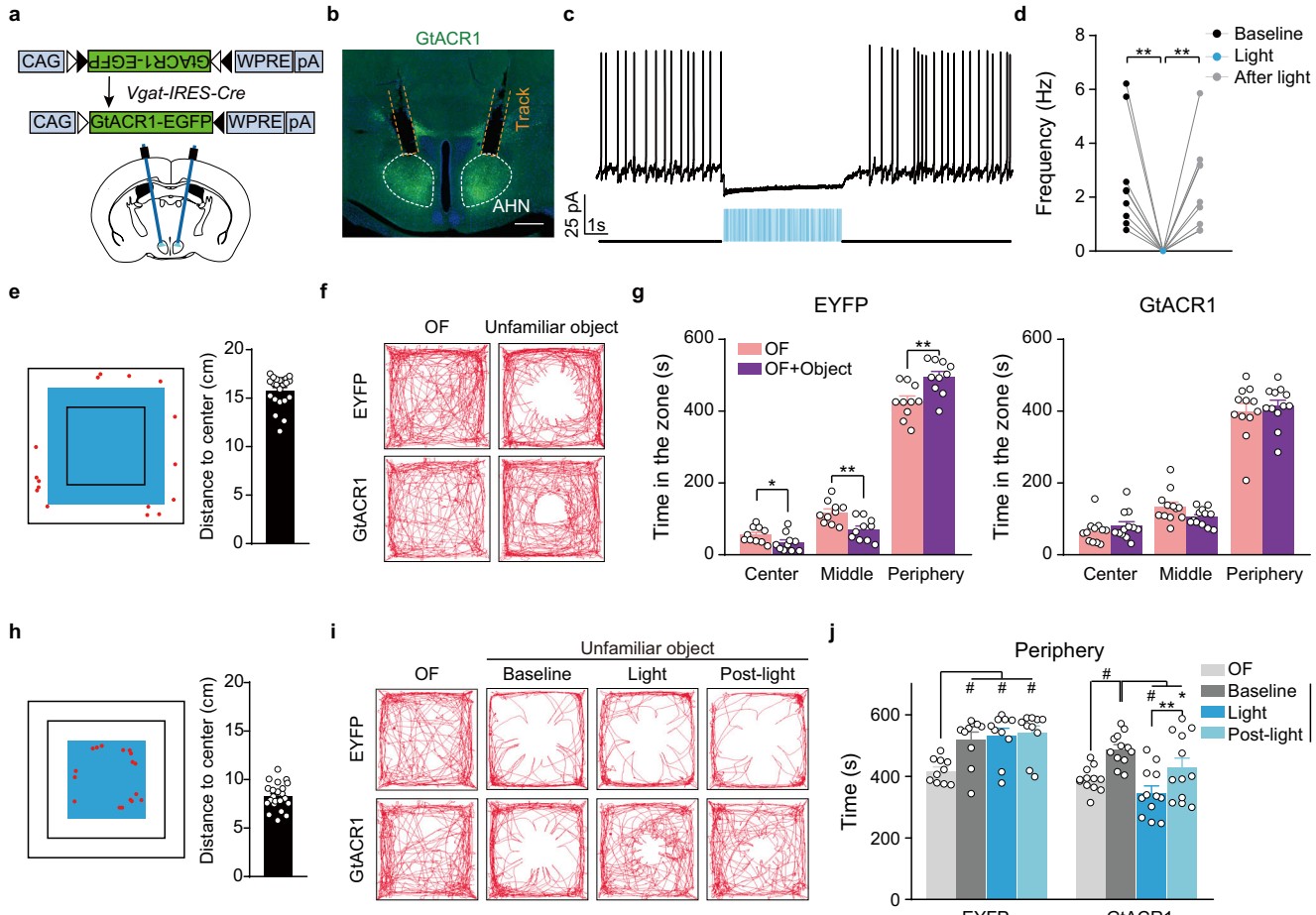

**Fig. 3 | Optogenetic inhibition of object-evoked AHN$^{Vgat+}$ activity reduces avoidance. a** The viral strategy to optogenetically inhibit AHN$^{Vgat+}$ neurons. **b** A representative post-hoc image showing GtACR1 expression in AHN and tracks of implanted fibers. Scale bar, 200 μm. **c**, **d** A representative trace (**c**) and quantifications (**d**) show light-mediated inhibition of GtACR1-expressing neurons. Wilcoxon matched-pairs signed rank test, baseline vs light, $p = 0.004$, light vs after light, $p = 0.004$. **e–j** Optogenetic inhibition of AHN$^{Vgat+}$ neurons. **e**, **h** Light delivery area shown by the blue square for experiments in **f**, **g** and **i**, **j**, respectively. Red dots denote the starting location of all approach (**e**) or retreat (**h**) bouts in a representative trial. The quantification on the right shows the average distance (in the vertical or horizontal direction) between the approach (**e**) or retreat (**h**) starting

location and the open field center, where the object was placed. $n = 22$ mice. **f**, **i** Representative trajectories of an EYFP or GtACR1 male with light delivered in the center and middle zone (**f**) or center zone only (**i**) after object introduction. **g**, **j** Quantification of the time spent in the indicated zone before or after object introduction in EYFP ($n = 10$) and GtACR1 males ($n = 12$). g. EYFP, two-tailed paired $t$ test, center, $p = 0.04$, middle, $p = 0.002$, peripheral, $p = 0.003$; GtACR1, center, Wilcoxon matched-paired t test, $p = 0.18$, middle and peripheral, two-tailed paired t test, $p = 0.08$ and 0.63, respectively. j. Two-way repeated measures ANOVA followed by Turkey's multiple comparison test. *$p < 0.05$; **$p < 0.01$; #, $p < 0.001$. Data are presented as mean values ±SEM.

---

activated ~20% of AHN$^{Vgat+}$ neurons, as shown by the amount of AHN$^{Vgat+}$ neurons co-expressing cytoplasmic *c-Fos*. Meanwhile, ~70–80% of nuclear-*c-Fos*-positive AHN$^{Vgat+}$ neurons co-expressed cytoplasmic *c-Fos* (Fig. 4c), indicating activation by both the first and the second stimulus. This convergence rate was comparable among all groups and was significantly higher than the chance level (~20%; Fig. 4c), suggesting that AHN$^{Vgat+}$ neurons activated by an unfamiliar object and fox urine overlap substantially.

To assess AHN neural responses with a better temporal resolution, we performed single-unit recordings in animals sequentially exposed to an unfamiliar object and fox urine (Supplementary Fig. 6d–g). We established via fiber photometry recordings that AHN$^{Vgat+}$ neurons as a population were most robustly activated during initial fox urine exposure (Supplementary Fig. 6a–c). Consistent with this result, out of 63 single-units recorded from three mice, thirteen tuned to initial fox urine exposure, and five to fox urine sniff. Independently, we identified nine single-units that tuned to the object approach, a neural ensemble that partially overlapped with fox urine-tuned single-units (Fig. 4d–f). Notably, the firing rates of urine-tuned single-units increased significantly during the object approach (Fig. 4g). On the other hand, the

object-tuned single-units, while showing the highest firing rates during the object approach (object approach, $7.5 \pm 1.8$ Hz; fox urine in, $4.2 \pm 1.3$ Hz; sniff, $5.8 \pm 1.7$ Hz; one-way repeated measures ANOVA summary, $F_{(1.433, 11.47)} = 9.037$, $p = 0.007$; Tukey's multiple comparisons, object approach vs. initial, $p = 0.02$, object approach vs. sniff, $p = 0.1$, initial vs. sniff, $p = 0.06$), did show a trend towards elevated activity during fox urine sniff (Fig. 4h). Together, these recording results show that AHN neural ensemble activated during the object approach overlapped with those responding to fox urine exposure. Combined with the catFISH data, these results support AHN$^{Vgat+}$ neuron activity as a shared mechanism between predator defense and anxiety.

## Progressive engagement of AHN$^{Vgat+}$ neurons during the elevated plus-maze test

Do AHN$^{Vgat+}$ neurons regulate anxiety levels and anxiety-like behaviors in other scenarios? To answer this question, we monitored the activity of AHN$^{Vgat+}$ neurons in mice exploring an elevated plus-maze (EPM), where avoidance of the open arm indicates general anxiety levels in the mice. In general, we found that the mouse exhibited significantly higher AHN$^{Vgat+}$ neuron activity in the open-arm than in

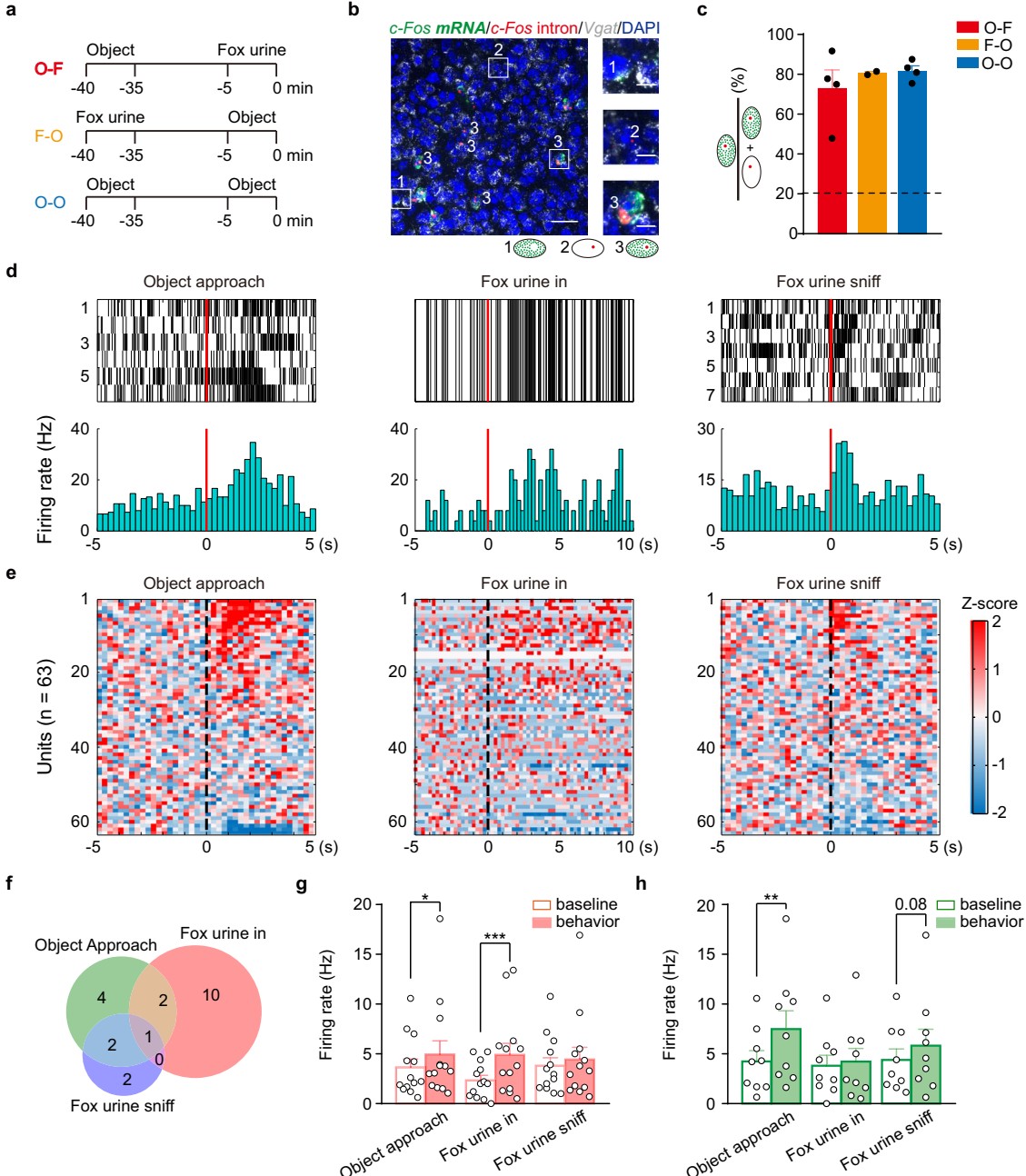

**Fig. 4 | Overlapping AHN neural ensembles respond to an object and a predator cue. a–c** catFish analysis of AHN$^{Vgat+}$ neurons activated by exposure to an unfamiliar object or fox urine. **a** Experimental design and the stimulus presentation order for each group. Animals were sacrificed at the time "0". **b** A representative image of an AHN brain section showing *Vgat*, *c-Fos* exon, and *c-Fos* intron in situ signals and their co-localization with DAPI as the counterstain. "1", "2", and "3" denote cells that only expressed *c-Fos* exon, or c-*Fos* intron, or both. Left, scale bar, 40 μm. Right, scale bar, 10 μm. **c** Convergence rate as measured by the percentage of *c-Fos*-intron-positive *Vgat* + neurons co-expressing *c-Fos* for each group. Each dot represents the value from an animal. The dashed line indicates the chance level. *n* = 4 O-F (object – fox urine), 2 F-O (fox urine – object) and 4 O-O (object – object). **d**–**h** Single-unit recordings of AHN neurons. **d** Raster plot (top) and the average (bottom) of the firing of an example single-unit aligned to the onset of object-approaching behavior

(left), fox urine in (middle), and fox urine sniff (right). **e** Heatmap representation of the normalized single-unit responses in Z scores sorted by response magnitude aligned to the behavioral onset, or the time of fox urine in at "0". *n* = 63 units from 3 mice. **f** Venn diagram showing single-units that tuned to "object approach" (green), "fox urine in" (pink), and "fox urine sniff" (purple). The number of single-units in each response category is indicated. **g, h** The average firing rates of the single-units tuned to initial fox urine in (g) and the object approach (**h**) during a specific epoch as indicated at the bottom. g. *n* = 13 units, Wilcoxon matched-pairs signed rank test, object approach, *p* = 0.03, fox urine in, *p* = 0.0002, fox urine sniff, *p* = 0.31. h. *n* = 9 units. Two-tailed paired t test, object approach, *p* = 0.006, fox urine in, *p* = 0.70, fox urine sniff, *p* = 0.08. *\*p* < 0.05; *\*\*p* < 0.01; *\*\*\*p* < 0.001. Data are presented as mean values ±SEM.

---

the closed arm, as shown by the heat map of recorded activity from an example mouse and quantification of the average GCaMP6s signal in the open and the closed arm (Fig. 5a, Δ*F/F*, open-arm, 1.5 ± 0.7%, closed arm, −0.4 ± 0.2%, *n* = 13 mice, Paired *t*-test, *p* = 0.048). No difference in signals was found for EYFP control mice (Δ*F/F*, open

arm, 0.4 ± 0.4%, closed arm, −0.2 ± 0.1%, *n* = 6 mice, Paired *t*-test, *p* = 0.29).

Mice exhibit progressive higher anxiety during repeated exposure to EPM[41]. This was supported by our observation of a marked reduction in open-arm exploration in the second trial compared to the first trial

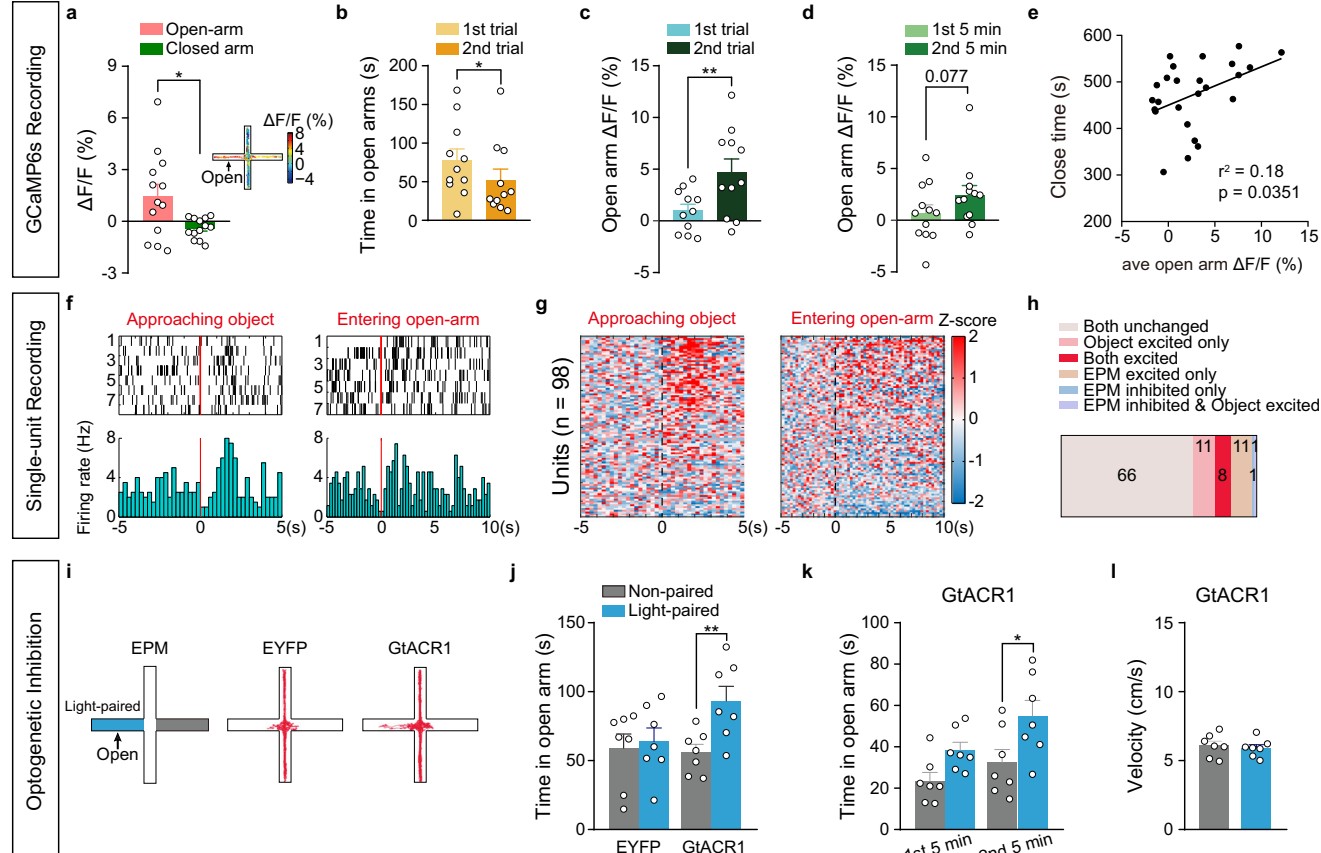

**Fig. 5 | Progressive engagement of AHN^Vgat+ neurons on EPM. a–e** Recording of AHN^Vgat+ GCaMP6s signals on EPM. **a** Average ΔF/F values in EPM open and closed arm for the first trial with the a heatmap representation of EPM ΔF/F value in an example (right). $n = 13$ mice. Two-tailed paired $t$ test $p = 0.048$. **b, c** Average open-arm time (**b**) and ΔF/F values (**c**) in the first or second trial. $n = 11$ mice. Open-arm time, Wilcoxon matched-pairs signed rank test, $p = 0.01$; ΔF/F, two-tailed paired $t$ test, $p = 0.006$. **d** Average ΔF/F values in the first or second 5 min of the first trial. $n = 12$ mice. **e** Correlation between average open-arm ΔF/F value and the time spent in the closed arm. $n = 14$ mice. Pearson correlation. **f–h** Single-unit recordings of AHN neurons. **f** Raster plot (top) and the average (bottom) of the firing rates of an example single-unit aligned to the onset of the object approach (left) and entering the EPM open-arm (right). **g** Heatmap representation of the normalized single-unit responses in Z scores sorted by response magnitude aligned to behavioral onset. $n = 98$ units from 3 mice. **h** Quantification of the number of single-units in each response category. **i–l** GtACR1-mediated optogenetic inhibition of AHN^Vgat+ neurons on EPM. **i** Left, schematics showing light delivery restricted to a random open-arm; right, example trajectories from an EYFP or a GtACR1 animal as indicated. **j, k** Time spent in EPM open-arm. **l** Velocity on EPM open-arm. $n = 7$ EYFP and 7 GtACR1 males. j. EYFP, Wilcoxon matched-pairs signed rank test, $p = 0.69$, GtACR1, two-tailed paired $t$ test, $p = 0.003$. k. Two-way repeated measures ANOVA followed by Sidak's multiple comparison test. *$p < 0.05$; **$p < 0.01$. Data are presented as mean values ±SEM.

(Fig. 5b). Notably, we found significantly higher AHN^Vgat+ activity in the open-arm during the second trial than during the first trial (Fig. 5c). A progressive increase in AHN^Vgat+ activity could be discerned even within the first trial, as shown by higher activity during the second 5 min than during the first 5 min of the trial (Fig. 5d). Similar progressive increase in AHN^Vgat+ activity was also detected during specific behaviors that occurred on the open-arm, including body elongation and head dipping (Supplementary Fig. 7a, b). Correlation analysis further showed that for all trials, the average open-arm GCaMP6s signals positively correlated with the total time the mouse spent in the closed arm (Fig. 5e), suggesting a connection between AHN^Vgat+ activity and anxiety levels on EPM.

Interestingly, the AHN^Vgat+ activity in the open-arm of EPM, as revealed by fiber photometry recordings, significantly correlated with object-evoked AHN^Vgat+ activity at the open field center ($r^2 = 0.30$, $p < 0.044$). To further assess this, we performed single-unit recording experiments in animals sequentially exposed to an unfamiliar object in the open field and the EPM (Supplementary Fig. 6d–f, h). The results showed that among the 20 single-unit channels that tuned to the object approach, eight also tuned to the EPM open-arm (Fig. 5f–h), accounting for 42% of all open-arm tuned single-units (Fig. 5f–h). This convergence rate was considerably higher than the chance level

(Fisher's exact test, $p = 0.074$). Thus, AHN neural responses on the EPM open-arm overlapped with those during the object approach. In other words, elevated AHN^Vgat+ neuron activity in the EPM open-arm may play a similar role as in object encounters to promote behavioral avoidance.

To test whether the activity of AHN^Vgat+ neurons is critical for EPM open-arm avoidance, we optogenetically inhibited these neurons by virally expressing GtACR1 in AHN^Vgat+ neurons and applied blue light via implanted optic fibers. The light was applied to only one of the two open-arms (Fig. 5i). We found that GtACR1 mice spent significantly more time exploring the light-illuminated open-arm while EYFP control mice spent a comparable amount of time in either open-arm (Fig. 5i, j). Interestingly, this behavioral effect appeared to be more substantial during the second 5 min than the first 5 min of the trial, consistent with the progressive increase of AHN^Vgat+ neuron activity described above (Fig. 5k). Importantly, locomotion was not affected by light stimulation (Fig. 5l). Furthermore, we found a similar increase in open-arm exploration in a new set of experiments when we shined the light to both open-arms to inhibit AHN^Vgat+ neurons (Supplementary Fig. 7c–f). Together, these results indicate that AHN^Vgat+ neuron activity reflects anxiety levels on EPM and is essential for EPM open-arm avoidance.

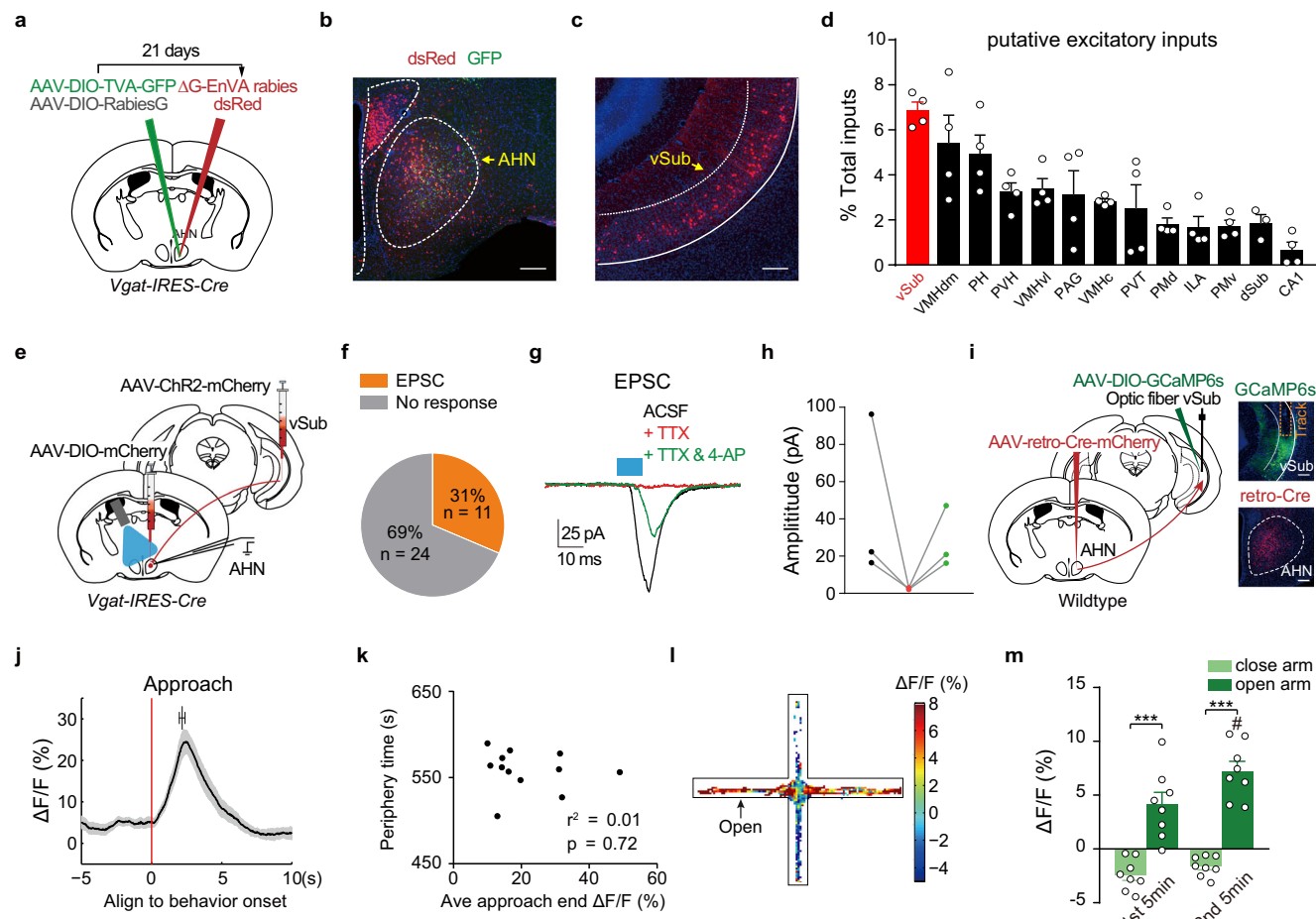

**Fig. 6 | Hippocampal formation sends monosynaptic excitatory inputs to AHN^Vgat+ neurons. a–d** Retrograde tracing of inputs to AHN^Vgat+ neurons. **a** Schematics of the viral strategy. **b, c** A representative image showing infection of AHN^Vgat+ neurons by AAV-DIO-TVA-GFP and EnVA-pseudotyped rabies virus expressing dsRed (**b**), and retrograde-labeled dsRed+ cells in vSub (**c**). Scale bar, 200 μm. **d** Quantification of dsRed+ neurons in candidate excitatory brain areas as the percentage of total dsRed+ cells detected outside AHN. Each circle represents data from one animal. $n = 4$ mice except for dSub. **e–h** Validation of vSub inputs to AHN^Vgat+ neurons as monosynaptic and excitatory via patch clamp. **e** Schematics of the viral and electrophysiological recording strategy to probe vSub inputs to AHN^Vgat+ neurons. **f** The number and percentage of recorded neurons that showed light-evoked EPSC. $n = 35$ cells from 6 animals. **g, h** Example traces (**g**) and quantifications (**h**) of light-evoked EPSC in AHN^Vgat+ neurons under different conditions. Blue bar indicating light pulse stimulation (10 ms). **i–m** Recording the activity of AHN-projecting vSub neurons. $n = 8$ mice. **i** Left, schematics of the viral strategy to target AHN-projecting vSub neurons retrogradely. Right, representative images showing GCaMP6s expressed in vSub and the track of the implanted fiber above (top), and retro-Cre expression in AHN (bottom). Scale bar, 200 μm. **j** Average GCaMP6s $\Delta F/F$ signals ±SEM (shades) in AHN-projecting vSub neurons aligned to approach onset. The bar on top shows the average time of retreat onset ±SEM. **k** No correlation between the average approach-end GCaMP6s $\Delta F/F$ value and the time animals spent in the periphery zone. $n = 12$ trials from 8 animals. Pearson correlation. **l** Heatmap depiction of EPM GCaMP6s $\Delta F/F$ signals in an example trial. **m** The average $\Delta F/F$ values in the closed arm and open-arm in the first and second 5 min of the trial. "#" denotes significant differences between open-arm $\Delta F/F$ values between the first and second 5 min ($p < 0.001$). Two-way repeated measures ANOVA followed by Sidak's multiple comparison test. ***$p < 0.001$. Data are presented as mean values ±SEM.

## Hippocampal formation sends monosynaptic excitatory inputs to AHN^Vgat+ neurons

We next sought to identify the synaptic inputs that drive AHN^Vgat+ neuron activity and avoidance behavior in anxiety-provoking situations, using a pseudorabies virus tracing strategy[42]. A mixture of AAVs encoding Cre-inducible avian retroviral receptor (TVA)-GFP and rabies glycoprotein (RG) was unilaterally injected into AHN of *Vgat-IRES-Cre* male mice, followed three weeks later with the injection of EnVA-coated pseudorabies virus expressing dsRed but lacking the glycoprotein into the same site (Fig. 6a). Our results showed many GFP +/dsRed+ "starter" cells in AHN (Fig. 6b) and retrograde-labeled dsRed + cells in many upstream brain regions (Fig. 6c and Supplementary Fig. 8a, b). For parallel controls, mice were injected with AAVs encoding Cre-inducible TVA but not RG (Supplementary Fig. 8c), a procedure preventing the spread of pseudorabies virus after infection of the "starter" cells. We found no dsRed+ cells in upstream brain

regions of these control mice (Supplementary Fig. 8c–e), validating the retrograde viral tracing strategy.

Quantification of labeled upstream neurons showed that AHN^Vgat+ neurons received significant inputs from the lateral septum (LS), medial preoptic area (MPO), and bed nucleus of stria terminalis (BNST) (Supplementary Fig. 8b), all of them are known to harbor predominantly GABAergic neurons (www.mouse.brain-map.org). On the other hand, among upstream regions likely to provide excitatory inputs to AHN^Vgat+ neurons, the ventral subiculum (vSub), which has been implicated in stress response, emotion regulation, and spatial navigation[43], had the highest percentage of retrogradely labeled neurons (Fig. 6d). vSub is part of the ventral hippocampal formation. It lies more posteriorly and is anatomically distinct from CA1 (Supplementary Fig. 9). We have identified far fewer retrogradely labeled neurons in CA1 or the dorsal subiculum (dSub) than vSub (Fig. 6d and Supplementary Fig. 9). Moreover, a vSub to AHN pathway has been

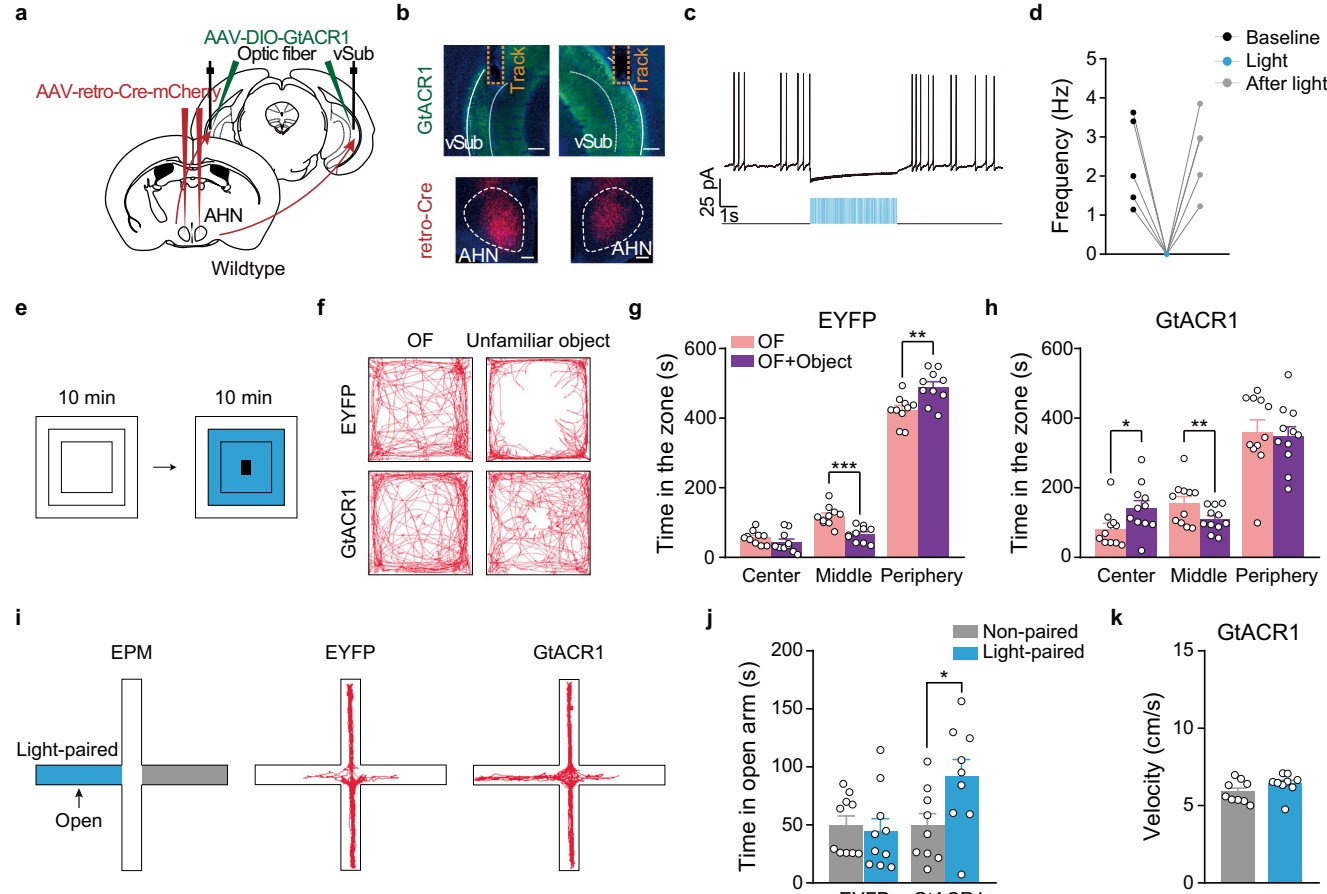

**Fig. 7 | Inhibiting AHN-projecting vSub neurons diminishes anxiety-like avoidance behavior. a** Schematics of the strategy to retrogradely target and bilaterally inhibit AHN-projecting vSub neurons. **b** Representative images showing GtACR1 expression in vSub (top), and retro-Cre expression in AHN (bottom). Scale bar, 200 μm. **c, d** A representative trace (**c**) and quantifications (**d**) showing trains of light pulses (473 nm, 20 ms, 20 Hz), represented as blue lines, acutely and reversibly inhibit the firing of GtACR1-expressing cells. **e–h** GtACR1-mediated inhibition of AHN-projecting vSub neurons in open field. *n* = 10 EYFP and 11 GtACR1 animals. **e** Schematics of light delivery restricted to the center and middle zone after object introduction. **f** Example open field trajectories of an EYFP and a GtACR1 male. **g, h** Time spent in the center, middle and periphery zones before and after object

introduction with light stimulation. **g** two-tailed paired t test, center, *p* = 0.24, middle, p < 0.0001, peripheral, *p* = 0.002. **h** Wilcoxon matched-pairs signed rank test for center, *p* = 0.04 and peripheral, *p* = 0.32 and two-tailed paired *t* test for middle, *p* = 0.006. **i–k** GtACR1-mediated inhibition of AHN-projecting vSub neurons on EPM. *n* = 10 EYFP and 9 GtACR1 animals. **i** Left, schematics showing light delivery restricted to one open-arm; right, example trajectories from an EYFP or a GtACR1 animal. **j** Time spent in the light-paired and non-paired open arm. EYFP, Wilcoxon matched-pairs signed rank test, *p* = 0.77, GtACR1, two-tailed paired t test, *p* = 0.03. **k** Velocity on EPM open-arm. Wilcoxon matched-pairs signed rank test. *p* < 0.05; **p* < 0.01; ***p* < 0.001. Data are presented as mean values ±SEM.

previously noted in other studies[39,44]. Together, these results show that vSub but not CA1 provides the principal hippocampal inputs to AHN[Vgat+] neurons.

To further confirm the monosynaptic connectivity from vSub neurons to AHN[Vgat+] neurons, we injected AAVs encoding hSyn-ChR2-mCherry unilaterally into vSub and AAVs encoding Cre-inducible mCherry into AHN of *Vgat-IRES-Cre* mice to label AHN[Vgat+] neurons fluorescently. We then performed patch-clamp recording from AHN[Vgat+] neurons in acute brain slices containing AHN to monitor synaptic activity evoked by vSub projections (Fig. 6e). Of 35 mCherry-expressing AHN[Vgat+] cells recorded from 6 mice, single light pulses (473 nm, 10 ms) evoked excitatory postsynaptic currents (EPSCs) in 11 cells (Fig. 6f–g), with a connection rate of 31%. The amplitude and latency for light-evoked EPSCs were 21.9 ± 7.8 pA and 5.5 ± 0.3 ms, respectively. Furthermore, tetrodotoxin (TTX) blocked light-evoked postsynaptic currents, which was reversed by the addition of 4-aminopyridine (4-AP) (Fig. 6g, h). Together, these results show the existence of monosynaptic excitatory inputs from vSub to AHN[Vgat+] neurons.

To examine whether AHN-projecting vSub neurons are responsible for AHN[Vgat+] neuron activation during anxiogenic situations, we

expressed GCaMP6s in vSub neurons projecting to AHN. This was achieved by injecting retroAAVs[45] encoding Cre-mCherry unilaterally into AHN of wildtype mice and AAVs encoding Cre-inducible GCaMP6s into vSub on the ipsilateral side (Fig. 6i). This approach labeled vSub neurons that projected to AHN, including those specifically targeting AHN[Vgat+] neurons. By monitoring GCaMP6s signals, we found that AHN-projecting vSub neurons displayed characteristic ramping activity during the approach-retreat bout in response to an unfamiliar object in the open field that aligned similarly to that found in AHN[Vgat+] neurons (Fig. 6j). However, unlike the AHN[Vgat+] neurons, the average approach-end GCaMP6s signals recorded from AHN-projecting vSub neurons did NOT correlate with the time animal spent in the peripheral zone away from the object (Fig. 6k, $r^2 = 0.01$, *p* = 0.72), suggesting that the activity of AHN[Vgat+] neurons tracked better with avoidance behaviors than AHN-projecting vSub neurons. Nevertheless, AHN-projecting vSub neurons showed higher and progressively increasing GCaMP6s signals in the open-arm than in the closed arm during the EPM test, similar to AHN[Vgat+] neurons (Fig. 6l, m). By comparison, when we recorded the activity of another population of neurons upstream of AHN, VMH *SF-1*-expressing neurons[33], we found no differences in GCamp6s signals

between EPM open and closed arm ($\Delta F/F$, open arm, $-0.23 \pm 0.22\%$, closed arm, $-0.005 \pm 0.08\%$, $n = 5$ mice, Paired t-test, $p = 0.44$). Taken together, the close correspondence between the activity patterns of AHN-projecting vSub neurons and AHN[Vgat+] neurons supports that vSub inputs drive AHN[Vgat+] neurons in anxiety-provoking situations.

## Inhibiting AHN-projecting vSub neurons diminishes anxiety-like avoidance behavior

Previous studies on ventral hippocampal regulation of anxiety behaviors almost exclusively targeted the ventral CA1 (vCA1)[13,46–54]. The AHN-projecting vSub neurons we identified are located posteriorly and anatomically distinct from vCA1 neurons (Supplementary Fig. 9). Unlike vCA1 neurons, vSub neurons have not been explicitly shown to regulate anxiety-like behaviors previously. To test whether AHN-projecting vSub neurons acutely regulate anxiety-like avoidance behavior, we specifically inhibited these neurons by bilaterally injecting retroAAVs encoding Cre-mCherry into AHN and AAVs encoding Cre-inducible GtACR1 into vSub (Fig. 7a, b). This approach labeled vSub neurons concreted in the region identified through pseudorabies viral tracing (Supplementary Fig. 10a; center of infection, Bregma, AP: $-4.19 \pm 0.02$ mm, $n = 11$ hit animals). We confirmed through ex vivo patch-clamp recordings that blue light pulses effectively and reversibly silenced GtACR1-expressing vSub neurons (Fig. 7c, d). By inhibiting the activity of AHN-projecting vSub neurons during the mouse's approach toward the unfamiliar object in the open field (Fig. 7e), we completely abolished object-induced avoidance behavior in GtACR1 animals (Fig. 7f–h). Critically, such inhibition had no effects on freezing, or jumping, with only a slight trend to reduce stretch attended postures (SAP; Supplementary Fig. 10b–e). As a control, in animals in which GtACR1 expression was not concentrated in the intended vSub target area, light delivery failed to affect object-induced center avoidance (Supplementary Fig. 10f). Additionally, inhibition of AHN-projecting vSub neurons significantly increased the time the mouse spent in the light-illuminated open-arm without affecting locomotion (Fig. 7i–k). Thus, these results demonstrate that the activity of AHN-projecting vSub neurons was required for anxiety-like avoidance behavior. It should be noted that AHN-projecting vSub neurons could regulate anxiety-like behaviors by affecting AHN[Vgat+] neurons and/or neurons in other downstream areas.

## Discussion

Despite extensive behavioral observation and theoretical work that connects predator defense and anxiety, the neural mechanisms that account for the convergence between the two were not well-understood. In this study, we reported overlapping activation patterns of AHN neurons in response to predator cue exposure and anxiogenic stimuli. Moreover, inhibiting AHN[Vgat+] neuron activity during exploration reduced anxiety-like avoidance behavior. We further identified vSub as a significant input from the hippocampal formation to AHN[Vgat+] neurons to drive avoidance behavior in anxiogenic situations. Together these results point to AHN[Vgat+] neurons as a site of convergence between predator defense and anxiety and highlight the evolutionary origin of anxiety-like emotions[4,5].

Curiously, unlike pan-neuronal activation of AHN, we failed to drive locomotion increases by optogenetically activating AHN[Vgat+] neurons (Supplementary Fig. 11), raising questions about whether AHN[Vgat+] neurons play a significant role in "post-encounter" defensive behaviors such as flight. Instead, our loss-of-function experiments unequivocally demonstrated the necessity of AHN[Vgat+] neuron activity for anxiety-like avoidance behavior. Furthermore, we found that AHN[Vgat+] neurons showed the most robust activation during initial exposure to predator cues. The activation profile of AHN[Vgat+] neurons in response to predator cues and their functional role in anxiety-like behaviors provide tentative experimental support for the Predator

Imminence Theory of anxiety, which proposes anxiety as the "pre-encounter mode" of defense.

Noteworthily, the exact identity of these anxiety-like neurons among the heterogeneous AHN[Vgat+] population remains to be determined. In particular, we have defined an anxiogenic function for AHN[Vgat+] neurons downstream of the vSub. However, distinct subsets of AHN[Vgat+] neurons may play different roles in the predator defense continuum and anxiety-like behaviors. Indeed, a recent study has shown that some AHN[Vgat+] neurons promote defensive attacks triggered by noxious mechanic stimuli mimicking predator bites[39]. Along this line, some AHN inhibitory neurons receiving inhibitory projections from lateral septum Crfr2-expressing neurons may inhibit stress-induced anxiety behaviors[12]. Thus, there may co-exist anxiolytic and anxiogenic AHN[Vgat+] neuronal populations representing subtypes of AHN neurons that serve opposite functions, analogous to the two subtypes of striatal medium spiny neurons expressing dopamine receptor 1 or 2[11,55]. In addition, AHN local circuits may link anxiolytic and anxiogenic neurons for modulating the approach vs. avoidance behavior in anxiety-provoking situations, similar to the local inhibitory microcircuits found in the central amygdala (CeA) for fear-related behaviors[56,57]. Detailed characterization of AHN[Vgat+] neurons in terms of transcriptional heterogeneity, activation patterns, and local- and long-range connectivity at single-cell resolution is of interest to further dissect the hypothalamic circuits for anxiety and predator defense.

Psychologists have long postulated that the hippocampal formation is a center for computing, comparing, and arbitrating "safety" and "threat" signals to coordinate approach vs. avoidance in anxiety-provoking situations[2,58,59]. Here, we identified vSub of the hippocampal formation as the primary hippocampal input to AHN[Vgat+] neurons. The AHN-projecting vSub neurons we identified are anatomically distinct from vCA1 neurons but are likely to receive direct inputs from vCA1[60]. We found that AHN-projecting vSub neurons showed progressively increasing activity on EPM, suggesting cumulation of internal regulatory signals during the behavior test. This vSub-to-AHN circuit may channel mental assessment of potential threats by the hippocampus and other cognitive brain areas to initiate motor programs for avoidance. Such a pathway would allow for flexible, context-dependent, and individually varied displays of anxiety-like avoidance behaviors.

Furthermore, our retrograde tracing study showed that AHN[Vgat+] neurons receive inputs from mPFC, LS, LH, vSub, and BNST, all of which are projection targets of vCA1 neurons. AHN[Vgat+] neurons may reside in a network position to integrate threat or safety-related signals transmitted and processed by these brain regions to initiate behavioral avoidance in anxiogenic situations. Importantly, we observed that AHN[Vgat+] neuron activity tracked closely with avoidance behavior rather than the type of threats, and its level of increase showed individual specificity across different test conditions. Thus, AHN[Vgat+] neurons may provide an entry point for understanding how excessive avoidance of perceived harm could emerge in some vulnerable individuals, such as psychiatric patients[61]. In short, our results offer insights into neural circuit mechanisms underlying the convergence between predator defense and anxiety and the emergence of individual characteristics in anxiety-like behavioral avoidance.

## Methods

### Animals

All animals used in the study were adult males aged between 8–30 weeks. Wildtype males of C57BL/6 J background were purchased from Shanghai SLAC Laboratory Animal Co., Ltd or Beijing Vital River Laboratory Animal Technology Co., Ltd. *Vgat-IRES-Cre* (*Slc32a1tm2(cre)Lowl/J*, Cat# 016962) and *Vglut2-IRES-Cre* (*Slc17a6tm2(cre)Lowl/J*, Cat# 016963) were purchased from Jackson Laboratory. The animals were housed with *ad libitum* food and

water under a reversed 12:12 h light-dark cycle with temperature controlled between 21 and 23 °C and humidity-controlled between 40 and 70% in the animal facility at the Institute of Neuroscience, except for those used in single-unit recording experiments and in Supplementary Fig. 6a (procedure 1), which were group-housed and bred in the animal facility at the Wuhan National Laboratory and National Institute of Biological Sciences respectively. Each cage contained at most six mice. Experiment protocols were approved by the Animal Care and Use Committee of the Institute of Neuroscience, Chinese Academy of Sciences, Shanghai, China (IACUC No. NA-01602016) or by the Hubei Provincial Animal Care and Use Committee and the Animal Experimentation Ethics Committee of Huazhong University of Science and Technology (IACUC No.844F) or by the Administrative Panel on Laboratory Animal Care at the National Institute of Biological Sciences.

## Virus

AAV-EF1α-DIO-mCherry (Serotype 2/8, titer $4.40 \times 10^{12}$ vg/mL, vector genome per mL), AAV-EF1α-DIO-ChR2-mCherry (Serotype 2/8, titer $9.39 \times 10^{12}$ vg/mL) and AAV-hSyn-ChR2-mCherry (Serotype 2/8, titer $8.40 \times 10^{12}$ vg/mL) were purchased from Obio Technology Co, Shanghai. AAV-CAG-DIO-GtACR1 (Serotype 2/8, titer $2.20 \times 10^{12}$ vg/mL) was purchased from Taitool Bioscience, Co, Shanghai. AAV-CAG-DIO-GtACR1 (Serotype 2/5, titer $5.00 \times 10^{12}$ vg/mL) was purchased from PackGene Biotech Co, Guangzhou. AAV-EF1α-DIO-H2B-EGFP (Serotype 2/8, titer $8.33 \times 10^{12}$ vg/mL), AAV-EF1α-DIO-EYFP (Serotype 2/8, titer $3.58 \times 10^{12}$ vg/mL), AAV-hSyn-DIO-GCaMP6s (Serotype 2/8, titer $4.80 \times 10^{13}$ vg/mL) and AAV-retro-hSyn-cre-mCherry (Serotype 2/2, titer $7.00 \times 10^{13}$ vg/mL) were purchased from gene editing core facility of Institute of Neuroscience. AAV-EF1α-DIO-RVG (Serotype 2/9, titer $2.00 \times 10^{12}$ vg/mL), AAV-EF1α-DIO-EGFP-2A-TVA (Serotype 2/9, titer $2.00 \times 10^{12}$ vg/mL) and RV-EnVA-DG-DsRed ($2.00 \times 10^8$ IFU/mL, infectious units per mL) were purchased from BrainVTA, Wuhan.

## Mouse surgery

Surgeries were performed as previously described[62]. Stereotaxic surgeries were performed on a David Kopf Model 1900 frame or a custom-built frame (Cat# SH-01, Xinglin LifeTech) that allows brain targeting at an angle. The animals were anesthetized with 0.8–5% isoflurane or with intraperitoneal (i.p.) injection of 1% pentobarbital sodium and hypodermic injection of 5 mg/kg carprofen for pain relief. The coordinates used for viral injection were based on the Paxinos and Franklin Mouse Brain Atlas, 2nd edition. For unilateral targeting of the AHN, coordinates of AP: −0.820 mm, ML: ± 0.500 mm, DV: −5.200 mm were used. For bilateral targeting of the AHN related to optogenetic inhibition experiments, the coordinates were adjusted to be AP: −0.820 mm, ML: ±1.400 mm, DV: −5.100 mm at an angle of 10 degrees. For targeting the vSub, the coordinates were AP: − 4.100 mm, ML: ± 3.650 mm, DV: −3.800 mm. ~ 60 − 200 nl of the virus was injected into the target brain site with a home-made nano-liter injector (Cat# SMO-10, Xinglin LifeTech) at a flow rate of ~ 70 nl/min. Optic fibers (diameter, 200 mm; N.A., 0.37; Hangzhou Newdoon Technology Co.,Ltd) were implanted ~50 μm above the viral injection site and secured onto the skull for fiber photometry recordings with dental cement and skull screws. For optogenetic inhibition, optic fiber was implanted 300−500 μm above the injection site. Animals were allowed to recover at least three weeks before being tested in behavioral experiments. For pseudorabies tracing experiment, ~80−150 nl of the 1:1 mixture of helper virus (AAV-DIO-TVA-GFP and AAV-DIO-RG), or ~ 100 nl AAV-DIO-TVA-GFP alone for control experiments, was first injected unilaterally in the AHN of *Vgat-IRES-Cre* mice and three weeks later, ~100−150 nl RV-EnVA-DG-DsRed into the exact location. Histological analysis was carried out 1 week later. Ovariectomized (OVX) surgeries were performed with animals anesthetized with i.p. injections of ketamine (80 mg/Kg) and xylazine (8 mg/Kg), and animals were allowed to

recover for over 1 week after the surgery prior to subsequent experiments.

## Histology

Histological analysis was performed as previously described[40,63]. Briefly, animals were anesthetized with 10% chloral hydrate and perfused with PBS, or DEPC treated PBS followed by 4% PFA. Brains were post-fixed overnight in 4% PFA at 4 °C and sectioned at 40 μm using a vibratome (VT1000S, Leica) except for experiments involving the RNAscope kit (ACD Bio.). All virally expressed fluorescent proteins or fusion proteins were visible without immunostaining. All brain sections processed for fluorescent staining were counterstained with DAPI (Sigma, Cat# d9542, 5 mg/ml, 1:1,000). Images were captured by a 10 X objective bright-field microscope (Nikon, Eclipse E600FN), or ×10 objective fluorescent microscope (Olympus, VS120) or confocal microscope (Nikon, C2). For pseudorabies virus tracing, brain sections were evenly divided into two sets, and only one set was mounted, imaged with a ×10 microscope (Olympus, VS120), and processed in ImageJ software. dsRed+ cells were counted outside the AHN injection site and assigned to specific brain areas according to the Allen Institute adult mouse coronal atlas (http://atlas.brain-map.org/). The percentage inputs (% inputs) were calculated for each injection site by dividing the number of dsRed+ cells found in each brain region or brain section by the total number of dsRed+ cells tallied.

For histological analysis involving the RNAscope kit, after perfusion and post-fix, brains were dehydrated with 30% sucrose in DPEC-PBS and sectioned at 20 μm using a microtome and mounted onto SuperFrost Plus® Slides (Fisher Scientific, Cat. No. 12-550-15). RNA probes for *Vgat* (Cat #319191, Cat #319191-C3), *Vglut2* (Cat #319171-C3), *c-Fos* intron (Cat #514521-C2), and *c-Fos* mRNA (Cat #316921) were ordered from ACD Bio. The in situ *hybridization* was performed using the RNAscope kit (ACD Bio.), following the user manual. Two brain sections covering the AHN were selected from each mouse. Images were captured with a 20X objective using a confocal microscope (Nikon C2) and processed in ImageJ software. Based on the DAPI counterstaining signal, the numbers of *Vgat*+ or *Vglut2*+ neurons in the AHN were counted. For catFISH experiments (Fig. 4b, c), after counting out AHN *Vgat*+ neurons, those that were either positive for *c-Fos* mRNA or intron or both were further tallied.

For validating the *Vgat-IRES-Cre* line, we used RNAscope Fluorescent Multiplex Assay combined with immune-fluorescent staining. One brain section was selected from each mouse. After the in situ, brain slices were blocked by 2.5% BSA (Sigma Cat #V900933) for an hour, then stained overnight at 4 °C with chicken anti-GFP antibody (ABCAM, Cat #ab13970, dilution 1:300). The next day, the brain sections were rinsed three times with 1 X PBS before incubating with the secondary antibody, goat-anti-chicken Alexa Fluor 488 (Jackson Immuno Research Laboratories, Cat #103-545-155, dilution 1:300) for two hours. Images were captured with 60X objective using a confocal microscope (Olympus FV3000) and processed in ImageJ software. Three 800 × 800-pixel squares were selected from each brain section, analyzed, and quantified for the proportion of co-labeled neurons.

For c-Fos fluorescent immunostaining, the brain was sectioned at 40 μm, and half of the brain sections covering AHN were blocked in 5% goat serum in AT (0.1% Triton and 2 mM MgCl₂ in PBS) for 1 h at room temperature, then incubated with guinea pig anti-c-Fos (Synaptic Systems, Cat #226004, dilution 1:1000) in AGT (0.5% goat serum, 0.1% Triton and 2 mM MgCl₂ in PBS) at 4 °C overnight. The next day, the brain sections were rinsed three times with AGT before incubating with the secondary antibody, goat-anti-guinea pig Alexa Fluor 647 (Jackson Immuno Research Laboratories, Cat #106-605-003, dilution 1:1000) for two hours. After several washes in AGT, AT, and PBS, brain sections were mounted onto glass slides. Images were captured with 20X objective using a confocal microscope (Nikon C2) and processed in ImageJ.

For c-Fos DAB staining, brain sections were similarly prepared as in fluorescent immunostaining. Half of the brain sections covering AHN were pretreated with 3% $H_2O_2$ at room temperature for 30 min and then washed twice in PBST (0.3% Triton X-100 in PBS). Next, brain sections were blocked with 5% goat serum in PBST at room temperature for 2 h and incubated with rabbit anti-c-Fos (Synaptic Systems, Cat #226003, dilution 1:20000) in PBST at 4 °C overnight. The next day, brain sections were rinsed six times with PBST and then incubated with the biotin-conjugated goat-anti-rabbit secondary antibody (Jackson Immuno Research Laboratories, Cat #111-065-003, dilution 1:1000) in PBST at room temperature for 2 h. After three washes in PBS, brain sections were stained with VECTASTAIN® ABC Reagent for 30 min following the manufacturer's manual. Following two washes in PBS, brain sections were incubated in 3,3-diaminobenzidine (Sigma, Cat #D5637-5G) solution with nickel intensification until the desired staining intensity was achieved. Then the reaction was stopped by rinsing sections in tap water. Brain sections were mounted onto glass slides. Images were captured with a ×10 objective using a microscope (Nikon, Eclipse E600FN) and processed in ImageJ.

## Behavioral tests

Mice were singly housed two days before behavioral experiments and were handled once per day for these two days. Animals were continuously singly housed during the period of behavioral tests. All behavior tests were recorded with a camera at a frame rate of 20, 25, or 30 Hz. For behavioral tests in the home cage, a stimulus, a hormonally primed ovariectomized female, was introduced after the animal was moved to the video-taping area to acclimate for ~10 min. For the open field (OF) test, mice were introduced into a corner of a $40 \times 40 \times 40$ cm white box under illumination, ~10 min after which, either an object (unfamiliar or familiar) was introduced into the OF or the experimenter's hand was put briefly about the box mimicking the motion of object introduction. Afterward, behaviors were recorded for another 10 min. The unfamiliar object used included a type C battery, a acrylic cuboid cube, a toy airplane, and a metal paper clip, presented on separate testing days in a pseudo-randomized manner. The familiar object used was a type C battery co-housed for three days with the tested animal. For behavioral analysis, the OF box was divided into three zones. The "center" zone encompasses the innermost $20 \times 20$ cm square; the "peripheral" zone is the region within 5 cm along the wall, and the rest is the "middle" zone. The EPM (Elevated Plus-Maze) apparatus used consists of a central region ($5 \times 5$ cm), two open-arms ($30 \times 5$ cm), and two close-arms ($30 \times 5 \times 15$ cm), in a "+" configuration and placed 50 cm above the floor. At the beginning of the EPM test, the mice were put in the center area oriented towards a close-arm.

To record AHN$^{Vgat+}$ neuron activity in response to fox urine exposure via fiber photometry, we first introduced the animal to a new clean cage for ~10 min for acclimation. Then, a piece of filter paper (semicircular, 7 cm diameter) was put into the cage, or 400–700 μl of saline was spotted onto the bedding as the control stimulus. The animal interacted with the control stimulus for 3–5 min, after which we swapped the clean filter paper with another one spotted with 400–700 μl of red fox urine (Lenonlures company, USA) or spotted the urine directly onto the bedding as the urine stimulus. The animal was recorded in the presence of fox urine for another 3–5 min.

To perform the single-unit recording of AHN neurons in response to an unfamiliar object and fox urine or EPM, we introduced a mouse into an open field arena and allowed it first to explore the arena for ~5 min. Afterward, an unfamiliar object was introduced into the center, and the mouse was monitored for another 5–10 min. Next, the mouse was introduced to a clean cage ($30 \times 20 \times 20$ cm) and allowed to explore for ~5 min. Then a semicircular filter paper (7 cm diameter) spotted with ~400 μL of red fox urine was introduced to one side of the

cage, and the mouse was monitored for another 5–10 min. Alternatively, the mouse was introduced on an EPM after the open field test for another 10–20 min.

All behavioral videos were annotated with custom-written MATLAB code as previously described[40]. "Approach start" was when the animal began to move toward the object, and "approach end" as the animal retreated from the object, which was also the start of the retreat. Retreat end was scored when animals stopped moving. During behavioral tests with a female mouse, a "social investigation" was defined as nose-to-face and nose-to-body contact initiated by the male towards the female, "sniff" was defined as nose-to-urogenital contact, and "mount" was defined as the male placing its forelimbs on the back of the female and climbing on top. "Freezing" was defined as a behavior with no other movement except for breathing. "Jump" was defined as all four limbs leaving the floor and moving upward. "SAP" (stretch attend posture) was defined as body elongation while moving slowly toward the object in the open field test. By comparison, "body elongation" was scored for similar behavior on EPM open-arm. "Head dipping" was defined as heading down toward the floor on EPM open-arm. The time point for "stimulus in" (filter paper, saline, and red fox urine) was defined as the point when the experimenter's hand was out of the cage.

In addition, the total time that animals spent in each open field zone or EPM arm was extracted with EthoVision XT (Noldus) or custom-written MATLAB code. Example trajectories were generated in EthoVision XT (Noldus). For the open field and EPM test, we calculated the velocity as the total distance divided by the moving time extracted with EthoVision XT. For ChR2 experiments, the velocity was the total distance traveled in each stage (before, during, or after light delivery) divided by 30 s.

For catFISH experiments, animals were randomly assigned into three groups and presented sequentially with an unfamiliar object in the open field (~5 min) or fox urine in a new cage (~5 min), separated by 30 min. Animals were immediately sacrificed after the second stimulus and processed for histological analysis.

## Fiber photometry

Fiber photometry recordings were carried out as previously described[40]. Before the recording, the implanted optic fiber was connected to the recording device (Biolink Optics Technology Inc., Beijing) through an external optic fiber. Briefly, a 488 nm laser was reflected through a dichroic mirror (MD498, Thorlabs), and the fluorescence signal was passed through a bandpass filter (MF525-39, Thorlabs) and collected in a photomultiplier tube (PMT, R3896, Hamamatsu). Emission signals were low-pass filtered at 30 Hz and sampled at 500 Hz with a data acquisition card (USB6009, National Instrument) using software provided by Biolink Optics. A LED bulb was transiently triggered at the start of the recording session to facilitate alignment of the fiber photometry recording signal and animal behaviors for data analysis.

For data analysis, fluorescent signals acquired were analyzed with custom-written MATLAB code. Briefly, raw signals were first adjusted according to the overall trend to account for photobleaching. Afterward, the values of fluorescence signal change ($\Delta F/F$) were calculated as $(F - F_0)/F_0$. In this formula, $F$ represents the signal value at any given moment, and $F_0$ represents the baseline. For recordings done in the open field, $F_0$ was the average signal value over the 10 min after the animals were placed in the open field and before the object introduction. For recording during fox urine exposure, $F_0$ was the average fluorescence value over the 10 min when the animals explored the new clean cage without introducing any control or experimental stimulus. When tested in the home cage, $F_0$ was the average fluorescence value over 10 min before introducing the stimulus. For the EPM test, $F_0$ was the mean fluorescence value for the 10 min recording period. To calculate the $\Delta F/F$ value for a defined

open field zone or EPM location, we first extracted the body location of the mice in each frame to assign the $\Delta F/F$ value to a specific zone or location and then averaged the $\Delta F/F$ values of all frames that belonged to a particular zone. To calculate the $\Delta F/F$ signal during fox urine exposure (filter paper, saline, and fox urine), we averaged the $\Delta F/F$ values for each 30 s window around the "stimulus in" time. To align $\Delta F/F$ signals with behavior, we segmented $\Delta F/F$ values based on behavior events and averaged first across different events in a trial and then across different trials from each animal.

To calculate the correlation between the GCaMP6s signal and approach-retreat bout, we first excluded the behaviors with an interbehavioral interval of less than one second. We then transferred the trend-adjusted F value and the behavior data into a binary (0 or 1) form and calculated the correlation between the two using a nonparametric Spearman correlation test. For GCaMP6s signal, we defined time points with an F value over two standard deviations (2 SD) away from the mean as "1" and otherwise as "0". For approach-retreat behavior, any time points annotated with the behavior was "1" and otherwise as "0". For all correlation analyses involving approach end $\Delta F/F$ signal, we excluded recordings of the first approach from the analysis to rule out any possible effects of initial exposure. We used a nonparametric Spearman correlation test to calculate the correlation between the approach end $\Delta F/F$ signal and the approaching interval in Fig. 1j. Other correlation analysis of $\Delta F/F$ signal and behavioral time were calculated using the parametric Pearson correlation test. Heatmap representations of $\Delta F/F$ value on EPM were generated with custom-written MATLAB code.

### Single-unit recording

Single-unit recording was performed and analyzed as previously[64,65]. Briefly, the guide tubes housed 16-channel electrodes of 25.4-mm formvar-insulated nichrome wire (Cat # 761500, A-M System, USA). The final impedance of the electrodes was 700–800kU. Mice were implanted with the 16-channel electrodes targeting AHN and then allowed to recover for at least five days before further behavioral tests. Before the testing, mice were singly housed and connected to the recording connector for two days to adapt. During the recording, the 16-channel electrodes were connected to an amplifier and sampled by a computer. Recorded signals were amplified (3200,000 gain) and digitized at 40 kHz by the NeuroPhys Acquisition System (Neurosys 2.8.0.8, USA) and NeuroLego System (Jiangsu Brain Medical Technology Co.ltd). Raw signals were filtered (300–6000 Hz) to remove field potential signals. Single-unit spike sorting was performed using the MATLAB toolbox (MClust-4.4). Waveforms with amplitudes smaller than 50–60 uV (three times the noise band) were excluded from the analysis. Unsorted waveforms were analyzed with peak value and two types of principal components. We manually defined waveforms with similar characters into clusters. A cluster of waveforms was considered a single neuron if the ratio of its inter-spike-interval (ISI) under 2 ms was less than 1%, the isolation distance was greater than 20, and L-ratio >0.1[66,67]. In addition, if the spike time of any two units coincided via the cross-correlation comparison, those units were also considered a single-unit.

The firing rate was analyzed by extracting the spike train frequency around a specific behavior or event. Data were binned by 250 ms. Neuron responses were calculated as $Z$-scores by normalizing the neural firing rates after the behavior or event onset to the firing rates before. Neurons with $Z$-score > 2 (p < 0.05) during any two consecutive bins were classified as excited neurons, whereas neurons with $Z$-score < −2 (p < 0.05) were classified as inhibited neurons. To analyze neural dynamics during the object approach and fox urine sniff, we extracted the single-unit firing rate 5 s before and after the onset of the behavior. To analyze neural dynamics around "fox urine in" or "entering an open-arm" events, we extracted the single-unit firing rate 5 s before and 10 s after the event. To judge whether a neuron was excited or inhibited during the object approach and fox urine sniff, we

considered the $Z$-score within 2 s of the behavioral onset. To judge whether a neuron was excited or inhibited after "fox urine in" or "entering the open-arm", we considered the $Z$-score within 10 s or 7.5 s of the event, respectively.

### Optogenetic inhibition

Before the test, the bilateral optic fibers were connected to a 473 nm laser power source (Shanghai Laser and Optics Century Co. or Changchun New Industries Optoelectronics Tech Co., Ltd.). Light delivery was controlled by LabState (AniLab), which detects the centroid of the animal in real-time to trigger the laser or turn it off. In the OF test, the light was triggered when the centroid of the animal entered the center and middle zone immediately after object introduction or when the centroid of the animal entered the center zone starting 10 min after object introduction for 10 min. For the second scenario, animals were monitored for another 10 min after cessation of light stimulation as the "post-light" stage. For the EPM test, the light was triggered when the centroid of the animal entered one randomly selected open-arm immediately after the animal was placed on the EPM or when the centroid of the animal entered either of the two open-arms 10 min after the animal was placed on the EPM, for a duration of 10 min. For the second scenario, animals were monitored for another 10 min after cessation of light stimulation as the "post-light" stage. For real-time place preference, the apparatus used consists of two 17 × 17 cm chambers and a 5-cm-wide gap in between the two chambers. One chamber was black with a metal-rod floor, and the other chamber was white with a wire floor. The light was triggered whenever the centroid of an animal entered a randomly chosen light-paired chamber. Light power in all these experiments was 5 mW, 20 Hz, 20 ms.

### Optogenetic activation

Animals were tested in the home cage. Before the test, the unilateral optic fiber was connected to a 473 nm laser power source (Shanghai Laser and Optics Century Co. or Changchun New Industries Optoelectronics Tech Co., Ltd.). A custom-written MATLAB code controlled the delivery of light pulses (5 mW, 20 Hz, 20 ms). We delivered 5 × 30 s of light, spaced ~180–220 s apart in a trial. To analyze light-induced c-Fos expression, we delivered the same light pattern as in the behavioral test and sacrificed the animal 1 h afterward.

### Brain slice electrophysiological recording

Mice were anesthetized with isoflurane, perfused transcardially with ice-cold oxygenated (95% $O_2$/5% $CO_2$) high-sucrose solution (in mM, 2.5 KCl, 1.25 $NaH_2PO_4$, 2 $Na_2HPO_4$, 2 $MgSO_4$, 213 sucrose, 26 $NaHCO_3$). Brains were sectioned coronally at 250 μm using a vibratome (Leica, VT1200S) in an ice-cold oxygenated high-sucrose solution. Brain sections containing the AHN or vSub were incubated in artificial cerebrospinal fluid (in mM, 126 NaCl, 2.5 KCl, 1.25 $NaH_2PO_4$, 1.25 $Na_2HPO_4$, 2 $MgSO_4$, 10 Glucose, 26 $NaHCO_3$, 2 $CaCl_2$) at 34 °C for 1 h. The intracellular solution for recordings contains (in mM) 135 K-gluconate, 4 KCl, 10 HEPES, 10 sodium phosphocreatine, 4 Mg-ATP, 0.3 $Na_3$-GTP, and 0.5 biocytin (pH:7.2, 276 mOsm). Recording electrodes (3–5 MΩ, Borosilicate Glass, Sutter Instrument) were prepared by a micropipette puller (Sutter Instrument, model P97). For synaptic transmission recordings, repetitive single pulses of blue light (10 ms, power 12 mW/mm$^2$) were delivered onto the brain slice through a 40 X objective with an X-Cite LED light source (Lumen Dynamics). Cells were clamped at 0 mV for IPSC recording and at −70 mV for EPSC recording. To validate the mono-synaptic connections between vSub and AHN neurons, 1 μM of tetrodotoxin (TTX, absin, Cat# abs44200985a) and 1 mM 4-aminopyridine (4-AP, Alomone Labs, Cat# A-115) were sequentially added into the bath solution. To confirm the effects of neuronal inhibition by GtACR1, repetitive 20 Hz pulses of blue light (20 ms, power 7 mW/mm$^2$, interval 20 s) were delivered onto the AHN or vSub brain slice. Whole-cell recordings were performed using a MultiClamp700B

amplifier and Digi-data 1440 A interface (Molecular Devices). Data were recorded with Clampex 10.2 (Molecular Devices) and low-pass filtered at 2 kHz and sampled at 20 kHz under voltage clamp, while low-pass filtered at 10 kHz and sampled at 10 kHz under current clamp. All experiments were performed at 33 °C with a temperature controller (Warner, TC324B).

## Statistics and reproducibility

Statistical tests were analyzed with GraphPad Prism (GraphPad Software). We first analyzed the data distribution with the Shapiro-Wilk normality test for comparisons between two groups. Datasets that passed the normality test were analyzed with Student's $t$ test (two-tailed, paired, or unpaired); otherwise, we used the Wilcoxon matched-pairs signed-rank test for paired data and used the nonparametric Mann–Whitney $U$-test for unpaired data. For data comparisons of more than two groups, one-way or two-way repeated measures ANOVA was used. The statistics used for each comparison were listed in Supplementary Table 1. All data were plotted as mean ± standard error of the mean (SEM). *$p < 0.05$; **$p < 0.01$; ***$p < 0.001$.

We repeated the findings with three batches of mice for the results presented in Figs. 1d–e and 3b. For results shown in Fig. 4b, supplementary Fig. 4a, c, one batch of mice was used. For results presented in Figs. 6b–c, i, supplementary Figs. 8b, d–e, and 9a, we repeated the findings with two batches of mice. For results shown in Fig. 7b and supplementary Figs. 4a, c, we replicated the findings with four batches of mice.

## Reporting summary

Further information on research design is available in the Nature Portfolio Reporting Summary linked to this article.

## Data availability

All data and material are available upon publication with a request. The source data used to generate the figures were deposited in http://github.com/xulab2022/AHN. Source data are provided with this paper.

## Code availability

We have deposited computer codes used in this project at https://github.com/xulab2022/AHN.

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

## Acknowledgements

We thank the Drs. Mu-ming Poo, Xiao-Ke Chen, Ning-Long Xu, Zhe Zhang, Ji Hu for their valuable comments on the manuscript, and Dr. Qing-Feng Wu for providing us AHN single-cell transcriptional data used in response to the reviewer. This work was supported by grants from the Ministry of Science and Technology of China (2021ZD0203203), the Strategic Priority Research Program of the Chinese Academy of Sciences (XDB32000000) and the National Nature Science Foundation of China (31871066, 31922028, 31900721), the Lingang Laboratory (LG202104-01-04), and by Shanghai Municipal Science and Technology Major Project (Grant No. 2018SHZDZX05).

## Author contributions

J.J.Y. and X.H.X. designed the experiments, analyzed the data and wrote the manuscript. X.J.D. performed behavioral tests and the catFISH experiments and analyzed the catFISH result. T.H. performed single-unit recordings with the help from M.H. under the supervision of H.L.; A.X.C. analyzed RV tracing, prepared the figures and scored behavioral videos. W.Z. performed electrophysiological recordings. Z.X.Y. performed *post-hoc* immunostaining of brain slices. X.Y.C. and Z.Y.X. performed part of fiber photometry experiments with fox urine under the supervision of P.C.; C.Y.W. analyzed videos under the supervision of C.X. Q.D.H. scored behavioral videos. X.Y.L. and X.Z. analyzed catFISH result. Y.L.Z. maintained the mouse colonies.

## Competing interests

The authors declare no competing interests.
