## [Peer Review File · Nature Communications]

A circuit from the ventral subiculum to anterior hypothalamic nucleus GABAergic neurons essential for anxiety-like behavioral avoidanceREVIEWER COMMENTS

Reviewer #1 (Remarks to the Author):

The authors performed an interesting study showing that activity in the projection from ventral subiculum to vgat anterior hypothalamus cells is sufficient and necessary for anxiety-related behaviors in mice. They also show increases in neural activity during proximity to open spaces in anterior hypothalamus vgat cells and in hypothalamus-projecting subiculum cells. Taken together, this work identifies a novel anxiety-inducing pathway using standard modern systems neuroscience tools. Overall the experiments are well-designed and executed efficiently. The methodology is sound is described appropriately. There are a few issues to be addressed prior to publication.

Major points

- 1. Show effect of all optogenetic manipulations on freezing, escape, jumps and stretch-attend postures in the open field.**
- 2. There are hundreds of reports from dozens of countries showing that mice and rats avoid the center of an open field, even after repeated exposures or long exposures such as 20 to 30 minutes. Thus, it is puzzling that the authors see very little open field center avoidance in the absence of the novel object. What explains the necessity of using a novel object to induce avoidance of center, when so many labs are able to reliably produce center avoidance without any novel objects?**
- 3. As the authors are aware, prior studies showed that activation of the projection from ventromedial hypothalamus to anterior hypothalamus causes flight. Thus, it is expected that activation of anterior hypothalamus cell bodies would also cause flight. Did the authors observe jumping/flight like motions following optogenetic activation of these cells?**
- 4. In figure 5D can the authors please add quantification of retrograde labelling in other hippocampal subfields?**
- 5. Please show effect of all optogenetic manipulations on overall locomotion speed.**
- 6. Please discuss prior recent work on hypothalamic predatory defense circuits**
<https://www.nature.com/articles/nn.3573.pdf>
<https://www.nature.com/articles/s41586-020-2728-4#Sec2>
<https://elifesciences.org/articles/69178>
<https://pubmed.ncbi.nlm.nih.gov/31917239/>
<https://pubmed.ncbi.nlm.nih.gov/33861942/>

Minor point

- 1. The anterior hypothalamus is a very long structure in the anterior-posterior axis. Can the authors perform additional experiments by activating more anterior and posterior locations in this structure to determine if the anxiety increasing effect happens specifically in a certain position or if excitation of this structure in any location produces increased anxiety?**

Reviewer #2 (Remarks to the Author):

The paper by Yan et al. examines how AHN vGAT+ neurons encode and control avoidance behavior in innately aversive contexts. The authors find that vGAT+AHN cells increase in activity during approach/retreat to/from a novel object in the OFT or to the open arms of the EPM, and not to control familiar objects or unfamiliar social interaction. They find that this effect correlates with increasing time in the EPM and

increasing time in the periphery of the OFT. They optogenetically inhibit these cells and find this increase diminishes. Using rabies tracing, they identify a projection from ventral subiculum to these AHN vGAT+ neurons and then silence those projecting vSub cells and see the same increase in open arm occupation. While the study of these anxiety circuits are of great interest, my primary concern is that results as they stand are not particularly novel, as described in my points below:

1) As cited by the authors, increased activity in the AHN has previously been reported in response to predator exposure eliciting avoidance behavior and optogenetic activation of inputs to the AHN can drive avoidance behavior. In addition, anxiolytics can reduce AHN activity, establishing a role of the AHN in the understood model of hypothalamic avoidance signaling. Thus the primary finding here is that, in object avoidance, it is the AHN vGAT+ neurons that are recruited. However, these cells are overwhelming the vast majority of cells in this region (83% as reported by authors) and experiments related to specificity of responses to these neurons are not provided. Further, previous work has already established an excitatory long range projection from the vSub to the AHN (Ding et al., 2020), as the authors show with rabies virus tracing experiments, and a number of studies have shown that ventral hippocampal cells projecting to the hypothalamus increase activity in the open arms of the EPM.

2) The authors suggest that the AHN is a site of convergence of the neural substrates of anxiety-related behaviors and predator defense behaviors. Outside of the unit recordings of AHN neurons during the unfamiliar object in the OFT and presentation of fox urine where only 3 of 63 neurons increased firing to both stimuli (not determined in the data if these are vGAT+ neurons), there is no clear data to support this. The object response may simply represent a novelty signal, rather than an anxiety signal as interpreted by the authors. Presentation of predator odor was not performed in the AHN vGAT+ photometry experiments but would have better indicated these cells' responsiveness in innate predator defense behavior as is discussed. To make the claim that anxiety-related and predator defense behaviors converge on cells in the AHN additional unit recordings showing a larger number of cells that are both responsive to the unfamiliar object approach and the open arm of the EPM would be needed.

3) The authors also incorrectly make the claim that their study is the first to show a progressive activation pattern of neurons involved in anxiety behaviors. First, while their photometry data shows an increase in open arm df/f of AHN-projecting cells in vSub in the second 5 mins compared to the first 5 mins of the EPM (Figure 5I), there is not a significant behavioral effect of optogenetic inhibition of these cells in the open arm when comparing the first or second 5 min periods (Figure 6K). In addition, vHPC-hypothalamic inhibition has been previously shown to only be effective after extended time in the EPM as the authors cite (Jimenez et al., 2018).

Other points:

Exact statistical tests should be provided for each panel. In some conditions (for example figure 4H), its not clear if these t-tests have been corrected or if a repeated measures ANOVA has been performed. As presented the statistical analyses cannot be evaluated.

Lines 209-212 its not clear why the EPM photometry data in initial exploration is not plotted, just described in the text. This seems like an important plot.

Some have argued that progressive reduction in open arm avoidance is not due to increased anxiety but rather less engagement in the task. Perhaps the authors could examine other anxiety related behaviors such as elongation or head dipping to increase understanding of how these neurons encode anxiety-related states.

Reviewer #3 (Remarks to the Author):

In this manuscript, the authors conduct a series of experiments aimed at linking anxiety-related behavior and predator defense. They focus their work on the role of the AHN, as this nucleus has been implicated in both classes of behavior. Using multiple behavioral assays, genetically-targeted loss of function approaches, in vivo recordings, retrograde tracing, and ephys, the authors identify a population of GABAergic neurons in the AHN as involved in the approach/retreat towards a novel object and anxiety-like behavior on an EPM. They further isolated a more specific role for vSub projections to these neurons in the control of these behaviors. The manuscript is well-written, the experiments are well designed and controlled for. The findings are exciting and help expand what we know about the connection between anxiety and predator defense. Technically, the manuscript is solid, however, I highlight a few fundamental conceptual issues which need to better be addressed and evaluated.

1) The main issue I have is a missed opportunity to really interrogate whether anxiety and acute predator defense behaviors sit on the same spectrum of a behavioral chain (as suggested by the authors when they reference models of predatory imminence), rather than simply being "linked" (and presumably engaging separable processes). Indeed, the most interesting potential for this paper is whether anxiety (as indexed here by reduced approach towards a novel object) represents the beginning stages of a predator defense continuum, and, whether behaviors at the opposite end of this this continuum (e.g. those elicited towards a predator) are controlled by the same population of cells. While the data presented reveal that an overlapping population of GABAergic neurons in the AHN are involved both in behavioral avoidance and response to predator urine, it is not clear whether they encode the entire predator defense continuum (which includes anxiety as an initial stage). While adequately addressing this would require a substantial increase in experiments, for the purpose of this manuscript, the authors could simply incorporate a more thorough discussion of predatory imminence and the potential role of these neurons in tracking the entire continuum vs. encoding anxiety and predator defense as separable (but overlapping) states.

2) The authors motivate some of their studies based on the fact that unfamiliar objects elicit behavioral changes that are "similar to those caused by brief predator odor" (line 92). While they cite one paper, there are plenty of papers demonstrate unique behaviors in response to a predator vs. a novel object. The authors should re-word this statement to take into account the differences between a predator and a novel object.

3) Related to point 2, Figure 2d demonstrates that while Vgat+ AHN neurons respond to both approach of a novel object and fox urine, the peak firing rate is lower for fox urine and more closely locked to/immediately following sniffing of urine. These suggest potentially distinct roles for this population in encoding these two types of behaviors. This should be addressed/explored through additional analyses of existing data (eg firing properties, rate, temporal dynamics during the entire repertoire of behaviors elicited in response to the cue) and/or through discussion in the text.

4) Line 187. Optogenetic inhibition of Vgat+ AHN neurons reduces object avoidance and this persists post-light delivery (Fig. 3i-j). Very interesting. Do the authors have a sense of why there effects are not time locked (persist)? Would be good to include in a sentence here.

5) The authors find that optogenetic inhibition of Vgat+ AHN neurons did not lead to place preference using a CPP task (Extended data Fig. 5c). Meanwhile, they find that the same manipulation increases time in the center of the OF+novel object (when thigmotaxis is usually observed). This results in the authors claiming their AHN population specifically encodes "object-induced anxiety" (line 200), which is an interesting finding (and central to their paper). However, it is still curious that a manipulation that is anxiolytic in one setting would not generalize to other settings, despite their differences in the type of anxiety produced. Moreover, this is further muddled by the fact that their AHN neurons show increased activity in the open arm of

an EPM (where no novel object is present), further suggesting a more general role for these neurons in overall anxiety. How do the authors square this? This section would benefit from some text addressing/further discussing this seeming paradox.

6) Experiments presented in Fig 5 are in WT mice and hence not specific to Vgat+ neurons in AHN which receive projections from vSub. The authors should point out that they can not rule out a role for Vglut+ neurons receiving vSub projections in their effects here (unless they have some Vglut data not shown here that they would like to include....).

7) Fig 6 highlights a role for AHN-projecting vSub neurons that mirrors some of the role seen for AHN Vgat+ neurons. This begs the question: if vSub effects upstream are similar to downstream effects in AHN, is AHN simply a relay station? What else does it add then?

Responses to the reviewer's comments

We thank all three reviewers for their thoughtful comments. We have performed new experiments and re-analyzed and re-plotted previous data to address the reviewer's concerns. We also re-organized and re-wrote parts of the manuscript to highlight the conceptual advance made in the current study.

Reviewer #1: *The authors performed an interesting study showing that activity in the projection from ventral subiculum to vgat anterior hypothalamus cells is sufficient and necessary for anxiety-related behaviors in mice. They also show increases in neural activity during proximity to open spaces in anterior hypothalamus vgat cells and in hypothalamus-projecting subiculum cells. Taken together, this work identifies a novel anxiety-inducing pathway using standard modern systems neuroscience tools. Overall the experiments are well-designed and executed efficiently. The methodology is sound is described appropriately. There are a few issues to be addressed prior to publication.*

We thank the reviewer for his or her thoughtful comments and the positive view of our work.

Major points

1. *Show effect of all optogenetic manipulations on freezing, escape, jumps and stretch-attend postures in the open field.*

In the revised manuscript, we analyzed the effects of optogenetic manipulations on freezing, jumps, and stretch-attend posture (SAP) and plotted the results in **Extended Data Fig. 3a-c**; **Extended Data Fig. 8a-c**. Optogenetic inhibition of AHN^{Vgat+} neurons mildly reduced freezing

without affecting the other two behaviors. We did not analyze escape because we found this behavior difficult to define objectively in our experiments.

2. There are hundreds of reports from dozens of countries showing that mice and rats avoid the center of an open field, even after repeated exposures or long exposures such as 20 to 30 minutes. Thus, it is puzzling that the authors see very little open field center avoidance in the absence of the novel object. What explains the necessity of using a novel object to induce avoidance of center, when so many labs are able to reliably produce center avoidance without any novel objects?

We have re-written this part of the text in the revised manuscript to clarify the point as the following: “Center avoidance and peripheral preference in an open field test are behavioral parameters that indicate rodent anxiety levels ⁶. As previously reported, we found wild-type male mice spent more time around the wall in an open field test (**Figure 1a**). However, by introducing an unfamiliar object (a battery) to the center of the open field ~10 mins after a mouse freely explored the arena, we found that this procedure led to more substantial center avoidance and peripheral preference (**Fig. 1a**), indicating that an unfamiliar object elevates the anxiety level of the animal.” Thus, we did see center avoidance in the open field without an object in the open field (**Fig. 1a-b, OF; Fig. 2c**). What we reported in the study was that the placement of the object further increased center avoidance compared to before object introduction, indicating that the object elevated the anxiety level of the animal.

3. As the authors are aware, prior studies showed that activation of the projection from ventromedial hypothalamus to anterior hypothalamus causes flight. Thus, it is expected that activation of anterior hypothalamus cell bodies would also cause flight. Did the authors observe jumping/flight like motions following optogenetic activation of these cells?

Previously, one study reported increased locomotion and jump behaviors elicited by pan-neuronal activation of AHN neurons¹. We also observed increased locomotion when pan-neuronally activating AHN neurons optogenetically. However, when we optogenetically activated AHN^{Vgat+} neurons specifically, we failed to detect any changes in movement velocity, indicating that activating AHN^{Vgat+} neurons is insufficient to drive locomotion increases as observed with pan-neuronal activation of AHN neurons. These data are plotted in the **Extended data Fig. 9** and discussed in the text as follows: “Curiously, unlike pan-neuronal activation of AHN, we failed to drive locomotion increases by optogenetically activating AHN^{Vgat+} neurons (Extended Data Fig. 9), raising questions about whether AHN^{Vgat+} neurons play a significant role in “post-encounter” defensive behaviors such as flight. Instead, our loss-of-function experiments unequivocally demonstrated the necessity of AHN^{Vgat+} neuron activity for anxiety-related avoidance behavior.”

4. In figure 5D can the authors please add quantification of retrograde labelling in other hippocampal subfields?

We have now quantified and plotted the percentage of inputs originating from other hippocampal subfields to AHN^{Vgat+} neurons in the updated **Fig. 6d** with corresponding changes made in the text.

5. Please show effect of all optogenetic manipulations on overall locomotion speed.

We have analyzed locomotion speed in all optogenetic manipulation experiments and plotted the data in corresponding figures. We found that light did not change locomotion speed.

6. Please discuss prior recent work on hypothalamic predatory defense circuits

<https://www.nature.com/articles/nn.3573.pdf>

<https://www.nature.com/articles/s41586-020-2728-4#Sec2>

<https://elifesciences.org/articles/69178>

<https://pubmed.ncbi.nlm.nih.gov/31917239/>

<https://pubmed.ncbi.nlm.nih.gov/33861942/>

We have expanded the section describing the hypothalamic predatory defense circuits in the introduction, incorporating the above list of references as follows: “The present study focused on the anterior hypothalamic nucleus (AHN), which reciprocally connects with the ventromedial hypothalamus (VMH) and the dorsal preammillary nucleus of the hypothalamus (PMd) to form the hypothalamus predator defense network in rodents ^{2,3}. Predator cues activate VMH and PMd ^{3–8}. Optogenetic activation of VMH neurons or their projections to AHN is sufficient to drive avoidance in mice ¹; however, loss-of-function experiments have not demonstrated a clear role for AHN in predator defense. By contrast, both gain- and loss-of-function experiments and studies of genetically defined populations have shown that VMH and PMd regulate essential post-encounter defense behaviors such as freezing and flight ^{9–13}.”

Minor point

1. The anterior hypothalamus is a very long structure in the anterior-posterior axis. Can the authors perform additional experiments by activating more anterior and posterior locations in this structure to determine if the anxiety increasing effect happens specifically in a certain position or if excitation of this structure in any location produces increased anxiety?

We thank the reviewer for raising this point. In addition to extending in the anterior-posterior axis, the anterior hypothalamus is presumably heterogeneous in neuronal types, connectivity, and function. Because of this complexity, we think it would be more productive first to characterize different AHN neuronal subtypes based on single-cell transcriptome profiling and

connectivity mapping and then to manipulate defined subpopulations of AHN neurons functionally. We discuss the AHN heterogeneity in the discussion section and will continue our research in this direction in the future.

Reviewer #2 (Remarks to the Author):

The paper by Yan et al. examines how AHN vGAT+ neurons encode and control avoidance behavior in innately aversive contexts. The authors find that vGAT+AHN cells increase in activity during approach/retreat to/from a novel object in the OFT or to the open arms of the EPM, and not to control familiar objects or unfamiliar social interaction. They find that this effect correlates with increasing time in the EPM and increasing time in the periphery of the OFT. They optogenetically inhibit these cells and find this increase diminishes. Using rabies tracing, they identify a projection from ventral fbiculum to these AHN vGAT+ neurons and then silence those projecting vSub cells and see the same increase in open arm occupation. While the study of these anxiety circuits are of great interest, my primary concern is that results as they stand are not particularly novel, as described in my points below:

We thank the reviewer for thinking that the study of these anxiety circuits is of great interest, with which we strongly agree. In the revised manuscript, we have performed new experiments, re-organized the figures, and re-worked the text to highlight the conceptual advance brought forth by our study. We believe these revisions have significantly strengthened the manuscript.

1) As cited by the authors, increased activity in the AHN has previously been reported in response to predator exposure eliciting avoidance behavior and optogenetic activation of inputs to the AHN can drive avoidance behavior. In addition, anxiolytics can reduce AHN activity, establishing a role of the AHN in the understood model of hypothalamic avoidance signaling. Thus the primary finding here is that, in object avoidance, it is the AHN vGAT+

neurons that are recruited. However, these cells are overwhelming the vast majority of cells in this region (83% as reported by authors) and experiments related to specificity of responses to these neurons are not provided. Further, previous work has already established an excitatory long range projection from the vSub to the AHN (Ding et al., 2020), as the authors show with rabies virus tracing experiments, and a number of studies have shown that ventral hippocampal cells projecting to the hypothalamus increase activity in the open arms of the EPM.

With all due respect, we strongly disagree with the notion that one could have deduced results reported in the manuscript from existing literature. In suggesting this, the reviewer might have confused several lines of evidence:

First, while pan-neuronal activation of AHN neurons promotes avoidance, this is **gain-of-function** evidence. One must perform **loss-of-function** experiments to demonstrate that AHN does participate or is essential for avoidance behaviors. Artificially stimulating many hypothalamic neuronal populations produce an avoidance phenotype in real-time place preference experiments. However, few are shown to regulate avoidance behavior with loss-of-function experiments. In the current work, we performed spatially and temporally resolved inhibition of AHN^{Vgat} neurons to prove that elevated AHN^{Vgat} neuron activity during exploration is specifically required for anxiety-related avoidance behaviors. To our knowledge, the current study provided the **first loss-of-function evidence** supporting the role of AHN in behavioral avoidance.

Secondly, we show in the revised manuscript that optogenetic activation of AHN^{Vgat+} neurons did not lead to locomotion increases as seen with pan-neuronal activation of AHN (Extended data Fig. 9), indicating that we should not equate pan-neuronal manipulation with cell-type

specific manipulation of the largest neuron population. These results highlight the importance of **cell-type specific functional dissections** as we did in the current study.

Thirdly, while in Ding et al., 2020 the authors reported that Sub neurons send axon projections to the anterior hypothalamus¹⁴, the authors did not validate this connection. Nor did they carry out any functional study of this pathway. In the current work, we performed retrograde virus tracing and *ex vivo* electrophysiological recordings to demonstrate that ~ 30% of AHN^{Vgat+} neurons received mono-synaptic excitatory inputs from vSub neurons. We further recorded the activity of and **functional manipulated** AHN-projecting vSub neurons to show that they regulate anxiety-related avoidance behavior. Our study represents the **first functional study** of this vSub → AHN pathway.

In addition, though extensively investigated, most studies of the ventral hippocampus (vHIP) in anxiety regulation focused on vCA1. To our knowledge, a role for vSub in anxiety-related behaviors has NEVER been demonstrated prior to our work using modern neuroscience tools. For example, in one of the most relevant references, titled “Anxiety Cells in a Hippocampal-Hypothalamic Circuit”, the authors reported that neurons in the vCA1 promoted EPM anxiety through projections to the lateral hypothalamus area (LHA). This vCA1 → LHA circuit differs entirely from this vSub → AHN pathway described in our study. Even though the phrase “hippocampal-hypothalamic Circuit” was used to describe both, it is inappropriate to equate the two pathways.

In this study, we reported overlapping activation patterns of AHN neurons in response to predator cue exposure and anxiogenic stimuli. We showed that inhibiting AHN^{Vgat+} neuron activity during exploration reduced anxiety-related avoidance behavior. We further identified

vSub as a significant input from the hippocampal formation to AHN^{Vgat+} neurons to drive avoidance behavior in anxiogenic situations. Together these results point to AHN^{Vgat+} neurons as a site of convergence between predator defense and anxiety and highlight the evolutionary origin of anxiety-related emotions. We believe this study is novel and will propel the field forward. More importantly, our assessment was met by comments from the other two reviewers, who have said, “*this work identifies a novel anxiety-inducing pathway using standard modern systems neuroscience tools*” (reviewer #1) and “*The findings are exciting and help expand what we know about the connection between anxiety and predator defense.*” (reviewer 3).

2) *The authors suggest that the AHN is a site of convergence of the neural substrates of anxiety-related behaviors and predator defense behaviors. Outside of the unit recordings of AHN neurons during the unfamiliar object in the OFT and presentation of fox urine where only 3 of 63 neurons increased firing to both stimuli (not determined in the data if these are vGAT+ neurons), there is no clear data to support this. The object response may simply represent a novelty signal, rather than an anxiety signal as interpreted by the authors. Presentation of predator odor was not performed in the AHN vGAT+ photometry experiments but would have better indicated these cells' responsiveness in innate predator defense behavior as is discussed. To make the claim that anxiety-related and predator defense behaviors converge on cells in the AHN additional unit recordings showing a larger number of cells that are both responsive to the unfamiliar object approach and the open arm of the EPM would be needed.*

We thank the reviewer for these critical comments. We have provided strong evidence **against** the notion that the object-evoked AHN^{Vgat+} response simply represents a novelty signal:

1. Object-evoked AHN^{Vgat+} activity showed little adaption as the trial continued when the novelty factor was expected to wear off after repeated encounters. This lack of adaption contrasts with recording results from another hypothalamic neuron population that detects and encodes the novelty of an object with a brief activity that rapidly adapted¹⁵.
2. Throughout the trial, AHN^{Vgat+} activity correlated with the latency to the following approach, indicating a correlation between AHN^{Vgat+} activity and anxiety-related avoidance.
3. A control experiment presenting a novel con-specific (a female) did not elicit changes in AHN^{Vgat+} activity. Thus, the object-evoked AHN^{Vgat+} response does not simply represent a novelty signal.

In addition, we have performed the following experiments and analyses in the revised manuscript to strengthen our conclusion that AHN is a site of convergence of the neural substrates underlying anxiety-related behaviors and predator defense behaviors:

1. We performed fiber photometry recordings of AHN^{Vgat+} neuron activity and showed that these neurons responded during fox urine exposure (**Extended data Fig. 5a-c**).
2. We performed the compartmental analysis of temporal activity by fluorescent in situ hybridization (catFISH) in animals sequentially exposed to an unfamiliar object and fox urine. We found that AHN^{Vgat+} neurons activated by either stimulus largely overlapped (**Fig. 4a-c**).
3. We re-analyzed the results of the single-unit recordings and identified 13 single-units that tuned to fox urine exposure. We found that the fire rates of these 13 single-units increased significantly during the object approach (**Fig. 4e-g**). Though not restricted to AHN^{Vgat+} neurons, these single-unit recording results combined with the catFISH data support that AHN^{Vgat+} neural ensemble activated during the object approach overlap with those responding to fox urine exposure.

4. we performed new single-unit recording experiments in animals sequentially exposed to an unfamiliar object and EPM. Out of 98 single-units recorded from 3 animals, 20 tuned to object approach and 19 tuned to EPM open-arm, while eight tuned to both. This convergent rate is above the chance level at $p = 0.074$.

We believe that adding these new pieces of evidence has strengthened our argument that AHN is a site of convergence for the neural substrates mediating anxiety-related behaviors and predator defense.

3) The authors also incorrectly make the claim that their study is the first to show a progressive activation pattern of neurons involved in anxiety behaviors. First, while their photometry data shows an increase in open arm df/f of AHN-projecting cells in vSub in the second 5 mins compared to the first 5 mins of the EPM (Figure 5I), there is not a significant behavioral effect of optogenetic inhibition of these cells in the open arm when comparing the first or second 5 min periods (Figure 6K). In addition, vHPC-hypothalamic inhibition has been previously shown to only be effective after extended time in the EPM as the authors cite (Jimenez et al., 2018).

We thank the reviewer for this critical comment. In the revised manuscript, we have removed the panel Fig. 6K and the sentence referred by the reviewer to avoid any overstatement.

Other points:

Exact statistical tests should be provided for each panel. In some conditions (for example figure 4H), its not clear if these t-tests have been corrected or if a repeated measures ANOVA has been performed. As presented the statistical analyses cannot be evaluated.

We have compiled details of the statistical methods for figure panels in the **Extended data Table 1**. All other statistical tests used in the result section were described along the way. Briefly, we first analyzed the data distribution with the Shapiro-Wilk normality test for comparisons between two groups. Datasets that passed the normality test were analyzed with Student's t-test (two-tailed, paired, or unpaired); otherwise, we used the Wilcoxon matched-pairs signed-rank test for paired data and used the nonparametric Mann–Whitney U-test for unpaired data. For data comparisons of more than two groups, one-way or two-way repeated measures ANOVA was used.

Lines 209-212 its not clear why the EPM photometry data in initial exploration is not plotted, just described in the text. This seems like an important plot.

We have modified the figure to include this result (**Fig. 5a**).

Some have argued that progressive reduction in open arm avoidance is not due to increased anxiety but rather less engagement in the task. Perhaps the authors could examine other anxiety related behaviors such as elongation or head dipping to increase understanding of how these neurons encode anxiety-related states.

Following the reviewer's advice, we have analyzed GCaMP6s signals in AHN^{Vgat+} neurons during specific behaviors, including elongation and head dipping. We found similar progressive increases in AHN^{Vgat+} neuron activity during the second trial than the first trial and the second 5 min than the first 5 min of the first trial (Extended data Fig. 6a-b). These results support that elevated AHN^{Vgat+} neuron activity reflects increased anxiety levels after repeated exposure to EPM.

Reviewer #3 (Remarks to the Author):

In this manuscript, the authors conduct a series of experiments aimed at linking anxiety-related behavior and predator defense. They focus their work on the role of the AHN, as this nucleus has been implicated in both classes of behavior. Using multiple behavioral assays, genetically-targeted loss of function approaches, in vivo recordings, retrograde tracing, and ephys, the authors identify a population of GABAergic neurons in the AHN as involved in the approach/retreat towards a novel object and anxiety-like behavior on an EPM. They further isolated a more specific role for vSub projections to these neurons in the control of these behaviors. The manuscript is well-written, the experiments are well designed and controlled for. The findings are exciting and help expand what we know about the connection between anxiety and predator defense. Technically, the manuscript is solid, however, I highlight a few fundamental conceptual issues which need to better be addressed and evaluated.

We thank the reviewer for his or her thoughtful comments and the positive view of our work. In the revised manuscript, we have performed new experiments, re-organized the figures, and re-worked the text to highlight the conceptual advance of the study. We believe these revisions have significantly strengthened the manuscript.

1) The main issue I have is a missed opportunity to really interrogate whether anxiety and acute predator defense behaviors sit on the same spectrum of a behavioral chain (as suggested by the authors when they reference models of predatory imminence), rather than simply being “linked” (and presumably engaging separable processes). Indeed, the most interesting potential for this paper is whether anxiety (as indexed here by reduced approach towards a novel object) represents the beginning stages of a predator defense continuum, and, whether behaviors at the opposite end of this this continuum (e.g. those elicited towards a predator) are controlled by the same population of cells. While the data presented reveal that an overlapping

population of GABAergic neurons in the AHN are involved both in behavioral avoidance and response to predator urine, it is not clear whether they encode the entire predator defense continuum (which includes anxiety as an initial stage). While adequately addressing this would require a substantial increase in experiments, for the purpose of this manuscript, the authors could simply incorporate a more thorough discussion of predatory imminence and the potential role of these neurons in tracking the entire continuum vs. encoding anxiety and predator defense as separable (but overlapping) states.

We thank the reviewer for this valuable comment. In the revised manuscript, we have added new evidence supporting anxiety as the “pre-encounter” mode of predator defense as suggested by the “Predator Imminence Continuum” theory. First, fiber photometry recordings of AHN^{Vgat+} neuron activity showed that these neurons responded more robustly during initial fox urine exposure than subsequent fox urine sniff (**Extended data Fig. 5a-c**). Secondly, we re-analyzed the results of the single-unit recordings and found that AHN neural ensemble activated during the object approach partially overlapped with those aroused by fox urine exposure (**Fig. 4d-h**). Thirdly, we found that optogenetically activating AHN^{Vgat+} neurons failed to drive locomotion increases seen with pan-neuronal activation of AHN, raising questions about whether these neurons play a significant role in “post-encounter” defensive behaviors such as flight. In addition, we added a more detailed description of the “Predator Imminence Continuum” theory in the introduction section. Furthermore, we extensively discussed the heterogeneity of AHN^{Vgat+} neurons and the possibility that different subpopulations may play distinct roles in anxiety and predator defense behaviors.

2) *The authors motivate some of their studies based on the fact that unfamiliar objects elicit behavioral changes that are “similar to those caused by brief predator odor” (line 92). While they cite one paper, there are plenty of papers demonstrate unique behaviors in response to a*

predator vs. a novel object. The authors should re-word this statement to take into account the differences between a predator and a novel object.

Following the reviewer's advice, we changed this sentence to "Although a still object differed greatly from a real predator, both produced a thigmotaxis phenotype in the open field test ⁹. We, therefore, inquired whether the hypothalamus predator defense circuit, particularly AHN, was engaged during object-induced center avoidance (**Fig. 1c**)."

3) *Related to point 2, Figure 2d demonstrates that while Vgat+ AHN neurons respond to both approach of a novel object and fox urine, the peak firing rate is lower for fox urine and more closely locked to/immediately following sniffing of urine. These suggest potentially distinct roles for this population in encoding these two types of behaviors. This should be addressed/explored through additional analyses of existing data (eg firing properties, rate, temporal dynamics during the entire repertoire of behaviors elicited in response to the cue) and/or through discussion in the text.*

We found via fiber photometry recordings that AHN^{Vgat+} neurons responded more robustly during initial fox urine exposure than subsequent fox urine sniff (**Extended data Fig. 5a-c**). Following the reviewer's advice, we analyzed the average fire rates of object-tuned neurons during initial fox urine exposure and fox urine sniff (**Fig. 4h**) and found the values to be much higher during the object approach than in response to fox urine. We did not perform other more complex analyses, such as "*temporal dynamics during the entire repertoire of behaviors elicited in response to the cue*" suggested by the reviewer. This is because, in these experiments, three mice were tested repeatedly to collect as many single-units as possible. While the present data did show convergence between object and predator cue-activated AHN neural ensembles, the experimental design and setup were not most appropriate for quantifying fox urine-induced behavioral changes. Future studies and more animal N numbers are needed

to more adequately characterize the response of AHN^{Vgat+} neurons during the entire repertoire of behaviors elicited by a predator cue or a natural predator.

4) *Line 187. Optogenetic inhibition of $Vgat+$ AHN neurons reduces object avoidance and this persists post-light delivery (Fig. 3i-j). Very interesting. Do the authors have a sense of why there effects are not time locked (persist)? Would be good to include in a sentence here.*

We thank the reviewer for raising this question. In the revised manuscript, we add new sentences to the paragraph to explain why we think this persistent effect happened, as follows: “In these experiments, the extended duration the mice spent in the presence of the object (30 min) did not reduce anxiety since EYFP control mice showed no reduction of object avoidance before and after light stimulation (**Fig. 3j**, Extended Data Fig. 3e-f). Instead, we found that GtACR1 animals spent more time close up to the unfamiliar object during light stimulation (Extended Data Fig. 3k). Some GtACR1 animals even climbed onto the object (Extended Data Fig. 3l), a behavior rarely observed in control males or before light stimulation in GtACR1 males. We suspect these close contacts fastened object familiarization, leading to reduced anxiety/avoidance after light termination”. The relevant figures were also updated.

5) The authors find that optogenetic inhibition of $Vgat+$ AHN neurons did not lead to place preference using a CPP task (Extended data Fig. 5c). Meanwhile, they find that the same manipulation increases time in the center of the OF+novel object (when thigmotaxis is usually observed). This results in the authors claiming their AHN population specifically encodes “object-induced anxiety” (line 200), which is an interesting finding (and central to their paper). However, it is still curious that a manipulation that is anxiolytic in one setting would not generalize to other settings, despite their differences in the type of anxiety produced. Moreover, this is further muddled by the fact that their AHN neurons show increased activity in the open

arm of an EPM (where no novel object is present), further suggesting a more general role for these neurons in overall anxiety. How do the authors square this? This section would benefit from some text addressing/further discussing this seeming paradox.

We thank the reviewer for raising this question. In the revised manuscript, we add the following sentences to clarify the point: “In a CPP apparatus consisting of two chambers (Extended Data Fig. 4a), EYFP or GtACR1 animals showed no consistent avoidance of either chamber (Extended Data Fig. 4b), indicating that neither is particularly anxiogenic. When we randomly paired light delivery to one of the two chambers (Extended Data Fig. 4c), light did not lead to a preference for the paired chamber in EYFP or GtACR1 animals (Extended Data Fig. 4d), demonstrating that inhibiting AHN^{Vgat+} neurons did not produce a CPP effect.” In other words, only in anxiogenic situations where AHN^{Vgat+} neurons presumably are activated will there be a behavioral effect of reduced avoidance when we inhibit AHN^{Vgat+} neurons. The relevant figure was also updated.

6) Experiments presented in Fig 5 are in WT mice and hence not specific to $Vgat+$ neurons in AHN which receive projections from vSub. The authors should point out that they can not rule out a role for $Vglut+$ neurons receiving vSub projections in their effects here (unless they have some $Vglut$ data not shown here that they would like to include....). add a disclaimer.

We thank the reviewer for raising this point. We have added the following two sentences in the result section as a disclaimer: “This approach labeled vSub neurons that projected to AHN, including those specifically targeting AHN^{Vgat+} neurons.”; “It should be noted that AHN-projecting vSub neurons may regulate anxiety-related behaviors via affecting AHN^{Vgat+} neurons, other AHN neurons, or neurons in other downstream targets.”

7) Fig 6 highlights a role for AHN-projecting vSub neurons that mirrors some of the role seen for AHN Vgat+ neurons. This begs the question: if vSub effects upstream are similar to downstream effects in AHN, is AHN simply a relay station? What else does it add then?

AHN. plot the difference, emphasize the discussion.

We thank the reviewer for raising this point. In the result section of the revised manuscript, we added the following sentences to describe the differences between the activity pattern of AHN^{Vgat+} neurons and AHN-projecting vSub neurons: “However, unlike the AHN^{Vgat+} neurons, the average approach-end GCaMP6s signals recorded from AHN-projecting vSub neurons did NOT correlate with the time animal spent in the peripheral zone away from the object (**Fig. 6k**, $r^2 = 0.01$, $p = 0.72$), suggesting that the activity of AHN^{Vgat+} neurons tracked better with avoidance behaviors than AHN-projecting vSub neurons.” We also extensively discussed the distinction between these two populations in regulating anxiety-related behaviors in the discussion section.

References:

1. Wang, L., Chen, I. Z. & Lin, D. Collateral pathways from the ventromedial hypothalamus mediate defensive behaviors. *Neuron* **85**, 1344–1358 (2015).
2. Canteras, N. S. The medial hypothalamic defensive system: hodological organization and functional implications. *Pharmacol Biochem Behav* **71**, 481–491 (2002).
3. Gross, C. T. & Canteras, N. S. The many paths to fear. *Nat Rev Neurosci* **13**, 651–658 (2012).
4. Canteras, N. S., Chiavegatto, S., Ribeiro do Valle, L. E. & Swanson, L. W. Severe reduction of rat defensive behavior to a predator by discrete hypothalamic chemical lesions. *Brain Res Bull* **44**, 297–305 (1997).

5. Cezario, A. F., Ribeiro-Barbosa, E. R., Baldo, M. V. C. & Canteras, N. S. Hypothalamic sites responding to predator threats--the role of the dorsal premammillary nucleus in unconditioned and conditioned antipredatory defensive behavior. *Eur J Neurosci* **28**, 1003–1015 (2008).
6. Papes, F., Logan, D. W. & Stowers, L. The vomeronasal organ mediates interspecies defensive behaviors through detection of protein pheromone homologs. *Cell* **141**, 692–703 (2010).
7. Carvalho, V. M. de A. *et al.* Representation of Olfactory Information in Organized Active Neural Ensembles in the Hypothalamus. *Cell Rep* **32**, 108061 (2020).
8. Mendes-Gomes, J. *et al.* Defensive behaviors and brain regional activation changes in rats confronting a snake. *Behavioural Brain Research* **381**, 112469 (2020).
9. Kennedy, A. *et al.* Stimulus-specific hypothalamic encoding of a persistent defensive state. *Nature* **586**, 730–734 (2020).
10. Kunwar, P. S. *et al.* Ventromedial hypothalamic neurons control a defensive emotion state. *Elife* **4**, (2015).
11. Silva, B. A. *et al.* Independent hypothalamic circuits for social and predator fear. *Nat Neurosci* **16**, 1731–1733 (2013).
12. Wang, W. *et al.* Coordination of escape and spatial navigation circuits orchestrates versatile flight from threats. *Neuron* **109**, 1848-1860.e8 (2021).
13. Wang, W. *et al.* Dorsal premammillary projection to periaqueductal gray controls escape vigor from innate and conditioned threats. *Elife* **10**, e69178 (2021).
14. Ding, S.-L. *et al.* Distinct Transcriptomic Cell Types and Neural Circuits of the Subiculum and Prosubiculum along the Dorsal-Ventral Axis. *Cell Reports* **31**, 107648 (2020).

15. Kosse, C. & Burdakov, D. Natural hypothalamic circuit dynamics underlying object memorization. *Nat Commun* **10**, 2505 (2019).

REVIEWER COMMENTS

Reviewer #1 (Remarks to the Author):

The authors have adequately addressed all my concerns and I support publication of the manuscript.

Reviewer #2 (Remarks to the Author):

In the first version of the manuscript, the main concern was related to the noteworthy nature of the result and the significant advance provided by this paper above previous work that has delineated a role for AHN in responses to predator odor, anxiety, and the generally well-understood model of AHN in hypothalamic avoidance signaling. New experiments provided do slightly better highlight the overlap between fox urine odor response and unfamiliar object response and EPM open-arm response overlap. And the statistical explanation helps clarify the tests used throughout the paper. The specific anxiety related behavior analyses also provide better evidence for AHN Vgat neurons have a clear role during anxiety like behavior. However, in this reviewers opinion, I am still not convinced by the arguments related to conceptual advance provided by these studies are sufficient for Nature Communications.

1) The authors state that two of the main advances are the optogenetic inhibition experiments and the slice electrophysiology. While this may constitute the first optogenetic inhibition experiment, similar experiments have been performed with optogenetic excitation. It is an interesting experiment and finding but doesn't substantially change our understanding that the AHN is involved in avoidance behavior. And yes, validating rabies tracing experiments with slice electrophysiology is crucial for substantiating a connection, a connection that has previously been reported using other methods.

2) The authors show that pan stimulation of AHN neurons leads to locomotion while vGAT+ stimulation does not, thus highlighting the specialized contribution of this cell type to behavior. However, this experiment does not get at this point. While the pan-neuronal vs. cell-type optogenetic activation experiment the authors perform is indicative that there might well be cell-type specific differences, vGAT cells are 83% of the neuronal population; showing that pan-neuronal activation produces locomotion, while vGAT optogenetic activation does not, is suggestive but not sufficiently demonstrative to nail down the specificity the authors are claiming. A more detailed analysis of the vGAT- population would be required.

3) The authors state that this is the first study to nail down the specificity of a vSUB-AHN pathway, while others were flawed in equating vCA1 and vSub. If we assume for a moment that this is the first paper that describes the vSub-AHN pathway, the question then becomes is that pathway specific to vSub-AHN. It is unclear in this study that the hippocampal to AHN pathway understudy is specific to vSUB. While the authors show histologically a greater concentration of cells from vSub to AHN, again that is insufficient to suggest a specific pathway. It's not clear in the paper how vSUB was differentiated from vCA1 and what markers were used to make this distinction. Also from figure 6D its not clear what part of CA1 is being analyzed.

Reviewer #3 (Remarks to the Author):

The authors adequately addressed all of my concerns.

Responses to the reviewer's comments

We are delighted that the previous revision has sufficiently addressed reviewers #1 and #3's concerns. We thank reviewer #2 for the thoughtful comments. We have included new results and re-plotted previous data in the revised manuscript. We sincerely hope these revisions will help repel the reviewer's concern about the manuscript.

Reviewer #2 (Remarks to the Author):

In the first version of the manuscript, the main concern was related to the noteworthy nature of the result and the significant advance provided by this paper above previous work that has delineated a role for AHN in responses to predator odor, anxiety, and the generally well-understood model of AHN in hypothalamic avoidance signaling. New experiments provided do slightly better highlight the overlap between fox urine odor response and unfamiliar object response and EPM open-arm response overlap. And the statistical explanation helps clarify the tests used throughout the paper. The specific anxiety related behavior analyses also provide better evidence for AHN Vgat neurons have a clear role during anxiety like behavior.

We are glad that the previous revision addressed many of the reviewer's concerns. In the revised manuscript, we have included new results from manipulating AHN Vglut2+ neurons to strengthen our claim about the specificity of AHN Vgat+ neuron function. In addition, we re-plotted the tracing and histological data to highlight the anatomical differences between vSub and vCA1 to support our claim on the specificity of the vSub to AHN connection in regulating anxiety behaviors. We hope that these new revisions will help address the reviewer's concerns.

However, in this reviewer's opinion, I am still not convinced by the arguments related to conceptual advance provided by these studies are sufficient for Nature Communications.

1) The authors state that two of the main advances are the optogenetic inhibition experiments and the slice electrophysiology. While this may constitute the first optogenetic inhibition experiment, similar experiments have been performed with optogenetic excitation. It is an interesting experiment and finding but doesn't substantially change our understanding that the AHN is involved in avoidance behavior. And yes, validating rabies tracing experiments with slice electrophysiology is crucial for substantiating a connection, a connection that has previously been reported using other methods.

To the reviewer's concerns about the presentation of only incremental advancement and not enough conceptual advance in the manuscript, we want first to make the philosophical argument that science, in general, is primarily an additive pursuit built on pre-existing knowledge and ideas. While ground-breaking discoveries and paradigm-shifting results are exciting, they are also rare. Rightfully so, as we develop our understanding of the world.

We built the manuscript on pre-existing theoretical work and behavioral data that suggest anxiety as an emotion is deeply rooted in predator defensive responses. We confirmed a pathway from vSub to AHN Vgat+ neurons and demonstrated its function *via* recording and optogenetic inhibition in controlling anxiety-related avoidance behavior. Though not emphasized in the title or the abstract, we showed that variations in AHN Vgat+ neuron activity could partly explain individual differences in anxiety-related behaviors. We believe the publication of these results will significantly propel the research on AHN function and deepen our understanding of anxiety and individuality in anxiety traits.

On the concern of whether the advancements mentioned above are enough to warrant a publication in *Nature Communications*, perhaps we can perform a hypothetical experiment. Let us pool twenty different papers of a similar genre previously submitted to *Nature Communications* and ask the reviewer to judge whether these papers should be accepted based on their conceptual advancement. It would be curious to see how many manuscripts deemed worthy or unworthy by the reviewer were accepted for publication. Could it be that the reviewer is holding our manuscript to a higher standard than the consensus? While we greatly appreciate the reviewer's high standard that has helped strengthen our manuscript along the revision process, we would also humbly ask for fairness in deciding the fate of the revised manuscript.

2) The authors show that pan stimulation of AHN neurons leads to locomotion while vGAT+ stimulation does not, thus highlighting the specialized contribution of this cell type to behavior. However, this experiment does not get at this point. While the pan-neuronal vs. cell-type optogenetic activation experiment the authors perform is indicative that there might well be cell-type specific differences, vGAT cells are 83% of the neuronal population; showing that pan-neuronal activation produces locomotion, while vGAT optogenetic activation does not, is suggestive but not sufficiently demonstrative to nail down the specificity the authors are claiming. A more detailed analysis of the vGAT- population would be required.

After further clarification on the point the handling editor relayed, we included results from the recording and functional manipulation of AHN Vglut2+ neurons in the revised manuscript (Extended data figure 4). The specificity of the Vglut2-Ires-Cre line and the validity of GtACR1-mediated optogenetic inhibition of these cells have been shown previously ¹. As shown below, while AHN Vglut2+ neurons were also activated in response to an object placed in the open field, optogenetic inhibition of these neurons during the object approach increased

object avoidance. In comparison, optogenetic inhibition of AHN Vgat+ neurons reduced object avoidance. Together, these results, in conjunction with the optogenetic experiment presented, support our claim that AHN Vgat+ neurons play a specific role in anxiety-related avoidance behavior.

In addition, as extensively discussed in the manuscript, we envision that AHN Vgat+ neurons are heterogeneous and can be further divided into different functional groups based on single-cell transcriptional profile and local or long-range connectivity. As shown in the Figure below, we performed a preliminary single-cell transcriptional profile of AHN and identified ten subtypes of Vgat+ neurons. We will investigate the function of different AHN Vgat+ neuron subtypes, but such a topic is beyond the scope of the current manuscript.

Figure 1. Single-cell transcriptome analysis of AHN Vgat+ neurons.

3) The authors state that this is the first study to nail down the specificity of a vSUB-AHN pathway, while others were flawed in equating vCA1 and vSub. If we assume for a moment that this is the first paper that describes the vSub-AHN pathway, the question then becomes is that pathway specific to vSub-AHN. It is unclear in this study that the hippocampal to AHN pathway

understudy is specific to vSUB. While the authors show histologically a greater concentration of cells from vSub to AHN, again that is insufficient to suggest a specific pathway. It's not clear in the paper how vSUB was differentiated from vCA1 and what markers were used to make this distinction. Also from figure 6D it's not clear what part of CA1 is being analyzed.

For Figure 6D, we analyzed all CA1 areas specified by the atlas. As stated in the method section, for quantification of RV tracing experiments, “dsRed+ cells were counted outside the AHN injection site and assigned to specific brain areas according to the Allen Institute adult mouse coronal atlas (<http://atlas.brain-map.org/>).” To more clearly elucidate this point, in the revised manuscript, we included a new supplementary Figure (Extended Data Figure 9). In this figure, we overlaid CA1, dSub, and vSub, as defined by the atlas, on top of the montage of brain sections from an example animal to show how we tallied the number of input cells in each brain area. It was easy to identify clustered dsRed+ cells in vSub but not CA1 or dSub. This observation was confirmed by the quantification shown in Figure 6D.

At the bottom of Extended Data Figure 9, we plotted the distribution of vSub inputs along the anterior-posterior axis from individual mice. One can see that vSub neurons projecting to AHN concentrated between Bregma -3.8mm to -4.3mm, much more posterior to vCA1. In fact, in all published work on ventral hippocampus regulation of anxiety behaviors, the coordinates used to target the ventral hippocampus were around Bregma, AP, -3.10mm. This coordinate is much more anterior to the coordinate we used to target vSub neurons that projected to AHN (Bregma, AP, -4.10mm). Thus, we can readily distinguish and specifically target vSub inputs to AHN based on the anatomical location along the AP axis.

In addition, we updated the Extended Data Figure 10 to show histological data of vSub GtACR1 expression in functional manipulation experiments. As expected, when we injected retroAAV into AHN, we labeled vSub neurons in a similar pattern as identified by RV tracing, shown in Extended Data Figure 10 by a montage of brain sections from an example animal. We also quantified that the center of GtACR1 expression was around Bregma, AP, 4.19 + 0.02mm, posterior to vCA1. More importantly, in miss-targeted animals which did not express strong GtACR1 within the targeted area, light delivery failed to reduce object-induced center avoidance in the open field. Together, these results support a vSub to AHN pathway specifically required for anxiety-related avoidance behavior.

REVIEWERS' COMMENTS

Reviewer #2 (Remarks to the Author):

the authors have addressed my concerns

Responses to the reviewer's comments

Reviewer #2 (Remarks to the Author):

the authors have addressed my concerns

We are delighted that the previous revision has sufficiently addressed reviewers #2's concerns.

We thank all three reviewers for their thoughtful comments.